# ROBUST GRAPH DICTIONARY LEARNING

**Weijie Liu,**[1,2] **Jiahao Xie,**[2] **Chao Zhang,**[2,8,9] **Makoto Yamada,**[3,4,5] **Nenggan Zheng,**[1,2,6,8] **Hui Qian**[2,7,8*]

[1] Qiushi Academy for Advanced Studies, Zhejiang University

[2] College of Computer Science and Technology, Zhejiang University

[3] Okinawa Institute of Science and Technology

[4] Kyoto University

[5] RIKEN AIP

[6] Zhejiang Lab

[7] State Key Lab of CAD&CG, Zhejiang University

[8] Alibaba-Zhejiang University Joint Research Institute of Frontier Technologies

[9] Advanced Technology Institute, Zhejiang University

`{westonhunter,xiejh,zczju,zng,qianhui}@zju.edu.cn,makoto.yamada@oist.jp`

## ABSTRACT

Traditional Dictionary Learning (DL) aims to approximate data vectors as sparse linear combinations of basis elements (atoms) and is widely used in machine learning, computer vision, and signal processing. To extend DL to graphs, Vincent-Cuaz et al. 2021 proposed a method, called GDL, which describes the topology of each graph with a *pairwise relation matrix* (PRM) and compares PRMs via the Gromov–Wasserstein Discrepancy (GWD). However, the lack of robustness often excludes GDL from a variety of real-world applications since GWD is sensitive to the structural noise in graphs. This paper proposes an improved graph dictionary learning algorithm based on a robust Gromov–Wasserstein discrepancy (RGWD) which has theoretically sound properties and an efficient numerical scheme. Based on such a discrepancy, our dictionary learning algorithm can learn atoms from noisy graph data. Experimental results demonstrate that our algorithm achieves good performance on both simulated and real-world datasets.

## 1 INTRODUCTION

Dictionary learning (DL) seeks to learn a set of basis elements (atoms) from data and approximates data samples by sparse linear combinations of these basis elements (Mallat, 1999; Mairal et al., 2009; Tošić and Frossard, 2011), which has numerous machine learning applications including dimensionality reduction (Feng et al., 2013; Wei et al., 2018), classification (Raina et al., 2007; Mairal et al., 2008), and clustering (Ramirez et al., 2010; Sprechmann and Sapiro, 2010), to name a few.

Although DL has received significant attention, it mostly focuses on vectorized data of the same dimension and is not amenable to graph data (Xu, 2020; Vincent-Cuaz et al., 2021; 2022). Many exciting machine learning tasks use graphs to capture complex structures (Backstrom and Leskovec, 2011; Sadreazami et al., 2017; Naderializadeh et al., 2020; Jin et al., 2017; Agrawal et al., 2018). DL for graphs is more challenging due to the lack of effective means to compare graphs. Specifically, evaluating the similarity between one observed graph and its approximation is difficult, since they are often with different numbers of nodes and the node correspondence across graphs is often unknown (Xu, 2020; Vincent-Cuaz et al., 2021).

The seminal work of Vincent-Cuaz et al. (2021) proposed a DL method for graphs based on the Gromov–Wasserstein Discrepancy (GWD) that is a variant of the Gromov–Wasserstein distance. Gromov–Wasserstein distance compares probability distributions supported on different metric spaces using pairwise distances (Mémoli, 2011). By expressing each graph as a probability measure and capturing the graph topology with a pairwise relation matrix (PRM), comparing graphs can be naturally formulated as computing the GWD, since both the node correspondence and the discrepancy of the compared graphs are calculated (Peyré et al., 2016; Xu et al., 2019b). However,

---

*Corresponding author.

observed graphs often contain structural noise including spurious or missing edges, which leads to the differences between the obtained PRMs and the true ones (Donnat et al., 2018; Xu et al., 2019b). Since GWD lacks robustness (Séjourné et al., 2021; Vincent-Cuaz et al., 2022; Tran et al., 2022), the inaccuracies of PRMs may severely affect GWD and the effectiveness of DL in real-world applications.

**Contributions.** To handle the inaccuracies of PRMs, this paper first proposes a novel robust Gromov–Wasserstein discrepancy (RGWD) which adopts a minimax formulation. We prove that the inner maximization problem has a closed-form solution and derive an efficient numerical scheme to approximate RGWD. Under suitable assumptions, such a numerical scheme is guaranteed to find a $\delta$-stationary solution within $\mathcal{O}(\frac{1}{\delta^2})$ iterations. We further prove that RGWD is lower bounded and the lower bound is achieved if and only if two graphs are isomorphic. Therefore, RGWD can be employed to compare graphs. RGWD also satisfies the triangle inequality which is of its own interest and allows numerous potential applications. A robust graph dictionary learning (RGDL) algorithm is thereby developed to learn atoms from noisy graph data, which assesses the quality of approximated graphs via RGWD. Numerical experiments on both synthetic and real-world datasets demonstrate that RGDL achieves good performance.

The rest of the paper is organized as follows. In Sec. 2, a comprehensive review of the background is given. Sec. 3 presents RGWD and the numerical approximation scheme for RGWD. RGDL is delineated in Sec. 4. Empirical results are demonstrated in Sec. 5. We finally discuss related work in Sec. 6.

## 2 PRELIMINARY

### 2.1 OPTIMAL TRANSPORT

We first present the notation used throughout this paper and then review the definition of the Gromov–Wasserstein distance that originates from the optimal transport theory (Villani, 2008; Peyre and Cuturi, 2018).

**Notation.** We use bold lowercase symbols (e.g. $\mathbf{x}$), bold uppercase letters (e.g. $\mathbf{A}$), uppercase calligraphic fonts (e.g. $\mathcal{X}$), and Greek letters (e.g. $\alpha$) to denote vectors, matrices, spaces (sets), and measures, respectively. $\mathbf{1}^d \in \mathbb{R}^d$ is a $d$-dimensional all-ones vector. $\Delta^d$ is the probability simplex with $d$ bins, namely the set of probability vectors $\Delta^d = \{\mathbf{a} \in \mathbb{R}_+^d \,|\, \sum_{i=1}^d a_i = 1\}$. $\mathbf{A}[i,:]$ and $\mathbf{A}[:,j]$ are the $i^{\text{th}}$ row and the $j^{\text{th}}$ column of matrix $\mathbf{A}$ respectively. Given a matrix $\mathbf{A}$, $\|\mathbf{A}\|_F$ and $\|\mathbf{A}\|_\infty$ denote its Frobenius norm and element-wise $\ell_\infty$-norm (i.e., $\|\mathbf{A}\|_\infty = \max_{i,j} |A_{ij}|$), respectively. The cardinality of set $\mathcal{A}$ is denoted by $|\mathcal{A}|$. The bracketed notation $[\![n]\!]$ is the shorthand for integer sets $\{1, 2, \dots, n\}$. A discrete measure $\alpha$ is denoted by $\alpha = \sum_{i=1}^m a_i \delta_{\mathbf{x}_i}$, where $\delta_{\mathbf{x}}$ is the Dirac measure at position $\mathbf{x}$, i.e., a unit of mass infinitely concentrated at $\mathbf{x}$.

**Gromov–Wasserstein distance.** Optimal Transport addresses the problem of transporting one probability measure towards another probability measure with the minimum cost (Villani, 2008; Peyre and Cuturi, 2018). The induced cost defines a distance between the two probability measures. Gromov–Wasserstein (GW) distance extends classic optimal transport to compare probability measures supported on different spaces (Mémoli, 2011). Let $(\mathcal{X}, d_\mathcal{X})$ and $(\mathcal{Y}, d_\mathcal{Y})$ be two metric spaces. Given two probability measures $\alpha = \sum_{i=1}^m p_i \delta_{\mathbf{x}_i}$ and $\beta = \sum_{i'=1}^n q_{i'} \delta_{\mathbf{y}_{i'}}$ where $\mathbf{x}_1, \mathbf{x}_2, \dots, \mathbf{x}_m \in \mathcal{X}$ and $\mathbf{y}_1, \mathbf{y}_2, \dots, \mathbf{y}_n \in \mathcal{Y}$, the $r$-GW distance between $\alpha$ and $\beta$ is defined as

$$\text{GW}_r(\alpha, \beta) := \left( \min_{\mathbf{T} \in \Pi(\mathbf{p}, \mathbf{q})} \sum_{i,j=1}^m \sum_{i',j'=1}^n D_{ii'jj'}^r T_{ii'} T_{jj'} \right)^{\frac{1}{r}},$$

where the feasible domain of the *transport plan* $\mathbf{T} = [T_{ii'}]$ is given by the set

$$\Pi(\mathbf{p}, \mathbf{q}) = \{\mathbf{T} \in \mathbb{R}_+^{m \times n} \,|\, \mathbf{T}\mathbf{1}^n = \mathbf{p}, \mathbf{T}^\top \mathbf{1}^m = \mathbf{q}\},$$

and $D_{ii'jj'}$ calculates the difference between pairwise distances, i.e., $D_{ii'jj'} = |d_\mathcal{X}(\mathbf{x}_i, \mathbf{x}_j) - d_\mathcal{Y}(\mathbf{y}_{i'}, \mathbf{y}_{j'})|$.

## 2.2 GRAPH REPRESENTATION AND COMPARISON

In this subsection, we formalize the idea of comparing graphs with GWD, which addresses the challenges that graphs are often with different numbers of nodes and the node correspondence is unknown (Xu et al., 2019b; Xu, 2020; Vincent-Cuaz et al., 2021).

**Pairwise relation and graph representation.**   Given a graph $\mathcal{G}$ with $n$ nodes, assigning each node an index $i \in [\![n]\!]$, $\mathcal{G}$ can be expressed as a tuple $(\mathbf{C}, \mathbf{p})$, where $\mathbf{C} \in \mathbb{R}^{n \times n}$ is a matrix encoding the pairwise relations (e.g. adjacency, shortest-path, Laplacian, or heat kernel) and $\mathbf{p} \in \Delta^n$ is a probability vector modeling the relative importance of nodes within the graph (Peyré et al., 2016; Xu et al., 2019b; Titouan et al., 2019; Vincent-Cuaz et al., 2022).

**Gromov–Wasserstein Discrepancy.**   GWD can be derived from the 2-GW distance by replacing the metrics with pairwise relations (Xu et al., 2019b; Vincent-Cuaz et al., 2022). More specifically, given an observed source graph $\mathcal{G}^{\text{s}}$ and a target graph $\mathcal{G}^{\text{t}}$ that can be expressed as $(\mathbf{C}^{\text{s}}, \mathbf{p}^{\text{s}})$ and $(\mathbf{C}^{\text{t}}, \mathbf{p}^{\text{t}})$ respectively, GWD is defined as

$$\text{GWD}\left((\mathbf{C}^{\text{s}}, \mathbf{p}^{\text{s}}), (\mathbf{C}^{\text{t}}, \mathbf{p}^{\text{t}})\right) = \left(\min_{\mathbf{T} \in \Pi(\mathbf{p}^{\text{s}}, \mathbf{p}^{\text{t}})} \sum_{i,j=1}^{n^{\text{s}}} \sum_{i',j'=1}^{n^{\text{t}}} (C_{ij}^{\text{s}} - C_{i'j'}^{\text{t}})^2 T_{ii'} T_{jj'}\right)^{\frac{1}{2}},$$

where $n^{\text{s}}$ and $n^{\text{t}}$ are the numbers of nodes of $\mathcal{G}^{\text{s}}$ and $\mathcal{G}^{\text{t}}$ respectively. GWD computes both a soft assignment matrix between the nodes of the two graphs and a notion of discrepancy between them. For conciseness, we abbreviate $\text{GWD}\left((\mathbf{C}^{\text{s}}, \mathbf{p}^{\text{s}}), (\mathbf{C}^{\text{t}}, \mathbf{p}^{\text{t}})\right)$ to $\text{GWD}(\mathbf{C}^{\text{s}}, \mathbf{C}^{\text{t}})$ in the sequel.

## 2.3 DICTIONARY LEARNING

Traditional DL approximates data vectors as sparse linear combinations of basis elements (atoms) (Mallat, 1999; Mairal et al., 2009; Tošić and Frossard, 2011; Jiang et al., 2015), and is usually formulated as

$$\min_{\mathbf{D} \in \mathcal{C}, \mathbf{W}} \sum_{k=1}^{K} \left\| \mathbf{X}[:, k] - \sum_{m=1}^{M} w_m^k \mathbf{D}[:, m] \right\|_2^2 + \lambda \Omega(\mathbf{w}^k), \tag{1}$$

where $\mathbf{X} \in \mathbb{R}^{d \times K}$ is the data matrix whose columns represent samples, the matrix $\mathbf{D} \in \mathbb{R}^{d \times M}$ contains $M$ atoms to learn and is constrained to the following set

$$\mathcal{C} = \{\mathbf{D} \in \mathbb{R}^{d \times M} | \forall m \in [\![M]\!], \|\mathbf{D}[:, m]\|_2 \leq 1\},$$

$\mathbf{W} \in \mathbb{R}^{M \times K}$ is the new representation of data whose $k^{\text{th}}$-column $\mathbf{w}^k = [w_m^k]_{m \in [\![M]\!]}$ stores the embedding of the $k^{\text{th}}$ sample, and $\lambda \Omega(\mathbf{w}^k)$ promotes the sparsity of $\mathbf{w}^k$. Such a formulation only applies to vectorized data.

Recently, Xu 2020 proposes to approximate graphs via the highly non-linear GW barycenter. Specifically, given a dataset of $K$ graphs which has PRMs $\{\mathbf{C}^k\}_{k \in [\![K]\!]}$ such that $\mathbf{C}^k \in \mathbb{R}^{n^k \times n^k}$, the basis elements $\{\bar{\mathbf{C}}^m\}_{m \in [\![M]\!]}$ are learned by solving

$$\min_{\{\bar{\mathbf{C}}^m\}_{m \in [\![M]\!]}, \{\mathbf{w}^k\}_{k \in [\![K]\!]}} \sum_{k=1}^{K} \text{GWD}^2\left(\mathbf{C}^k, \mathbf{B}\left(\mathbf{w}^k, \{\bar{\mathbf{C}}^m\}_{m \in [\![M]\!]}\right)\right),$$

where $\mathbf{w}^k \in \Delta^M$ is referred to as the embedding of the $k^{\text{th}}$ graph $\mathcal{G}^k$, and the GW barycenter $\mathbf{B}\left(\mathbf{w}^k, \{\bar{\mathbf{C}}^m\}_{m \in [\![M]\!]}\right)$ gives the approximation of $\mathcal{G}^k$ and is defined as

$$\mathbf{B}\left(\mathbf{w}^k, \{\bar{\mathbf{C}}^m\}_{m \in [\![M]\!]}\right) = \operatorname*{argmin}_{\mathbf{B}} \sum_{m=1}^{M} w_m^k \, \text{GWD}^2(\mathbf{B}, \bar{\mathbf{C}}^m).$$

Therefore, a complex bi-level optimization problem is involved, which is computationally inefficient (Vincent-Cuaz et al., 2021).

**DL for graphs.** To overcome the above computational issues, Vincent-Cuaz et al. 2021 proposed GDL which approximates each graph as a weighted sum of PRMs and is formulated as

$$\min_{\{\bar{\mathbf{C}}^m\}_{m \in [\![M]\!]}, \{\mathbf{w}^k\}_{k \in [\![K]\!]}} \sum_{k=1}^{K} \text{GWD}^2 \left( \mathbf{C}^k, \sum_{m=1}^{M} w_m^k \bar{\mathbf{C}}^m \right) + \lambda \Omega(\mathbf{w}^k),$$

where each atom $\bar{\mathbf{C}}^m$ is a $n^{\text{a}} \times n^{\text{a}}$ matrix. In contrast to the $\ell_2$ loss in Eq. (1), GWD is used to assess the quality of the linear representation $\sum_{m=1}^{M} w_m^k \bar{\mathbf{C}}^m$ for $k \in [\![K]\!]$. However, the observed graphs often contain noisy edges or miss some edges in real-world applications (Clauset et al., 2008; Xu et al., 2019b; Shi et al., 2019; Piccioli et al., 2022), which leads to the inaccuracies of the PRMs $\mathbf{C}^k$, that is, the deviation between $\mathbf{C}^k$ and the true PRM $\mathbf{C}^{k^*}$. Since GWD lacks robustness (Séjourné et al., 2021; Vincent-Cuaz et al., 2022; Tran et al., 2022), the quality of the learned dictionary may be severely affected.

## 3 ROBUST GROMOV–WASSERSTEIN DISCREPANCY

To deal with the inaccuracies of PRMs, this section defines a robust variant of GWD, referred to as RGWD. The properties of RGWD are rigorously analyzed. We then derive a theoretically guaranteed numerical scheme for calculating RGWD approximately. Due to the limit of space, all proofs can be found in the appendix.

**Definition 1** *Given an observed source graph $\mathcal{G}^{\text{s}}$ and a target graph $\mathcal{G}^{\text{t}}$ that can be expressed as $(\mathbf{C}^{\text{s}}, \mathbf{p}^{\text{s}})$ and $(\mathbf{C}^{\text{t}}, \mathbf{p}^{\text{t}})$ respectively, RGWD is defined by the solution to the following problem*

$$\text{RGWD}\left((\mathbf{C}^{\text{s}}, \mathbf{p}^{\text{s}}), (\mathbf{C}^{\text{t}}, \mathbf{p}^{\text{t}}), \epsilon\right) = \left( \min_{\mathbf{T} \in \Pi(\mathbf{p}^{\text{s}}, \mathbf{p}^{\text{t}})} \max_{\mathbf{E} \in \mathcal{U}_\epsilon} f(\mathbf{T}, \mathbf{E}; \mathbf{C}^{\text{s}}, \mathbf{C}^{\text{t}}) \right)^{\frac{1}{2}},$$

*where the objective $f(\cdot)$ is given by*

$$f(\mathbf{T}, \mathbf{E}; \mathbf{C}^{\text{s}}, \mathbf{C}^{\text{t}}) = \sum_{i,j=1}^{n^{\text{s}}} \sum_{i',j'=1}^{n^{\text{t}}} (C_{ij}^{\text{s}} - C_{i'j'}^{\text{t}} - E_{i'j'})^2 T_{ii'} T_{jj'},$$

*and the perturbation $\mathbf{E}$ is in the bounded set $\mathcal{U}_\epsilon = \{\mathbf{E} | \mathbf{E} = \mathbf{E}^\top \text{ and } \|\mathbf{E}\|_\infty \le \epsilon\}$.*

RGWD requires the sought transport plan to have low transportation costs for all perturbation $\mathbf{E}$ in set $\mathcal{U}_\epsilon$. For succinctness, we omit $\mathbf{C}^{\text{s}}$ and $\mathbf{C}^{\text{t}}$ in $f(\mathbf{T}, \mathbf{E}; \mathbf{C}^{\text{s}}, \mathbf{C}^{\text{t}})$ in the following.

### 3.1 PROPERTIES OF RGWD

The properties of RGWD are presented as follows. Firstly, although RGWD involves a non-convex non-concave minimax optimization problem, the inner maximization problem has a closed-form solution, which allows an efficient numerical scheme for RGWD. Secondly, RGWD has a lower bound that is achieved if and only if the expressions of compared graphs are identical up to a permutation, which implies RGWD can be employed to evaluate the similarity between one observed graph and its approximation in DL. Thirdly, RGWD satisfies the triangle inequality, which allows numerous potential applications including clustering (Elkan, 2003; HajKacem et al., 2019), metric learning (Pitis et al., 2019), and Bayesian learning (Moore, 2000; Xiao et al., 2019). Finally, arbitrarily changing the node orders does not affect the value of RGWD. More formally, we state the properties in the following theorem.

**Theorem 1** *Given an observed source graph $\mathcal{G}^{\text{s}}$ and a target graph $\mathcal{G}^{\text{t}}$ that can be expressed as $(\mathbf{C}^{\text{s}}, \mathbf{p}^{\text{s}})$ and $(\mathbf{C}^{\text{t}}, \mathbf{p}^{\text{t}})$ respectively, RGWD satisfies*

> *1. for all $\mathbf{T} \in \Pi(\mathbf{p}^{\text{s}}, \mathbf{p}^{\text{t}})$, $\mathbf{E}(\mathbf{T}) = [E_{i'j'}(\mathbf{T})]$ where*
>
> $$E_{i'j'}(\mathbf{T}) = \begin{cases} \epsilon, & \text{if } \sum_{i,j=1}^{n^{\text{s}}} T_{ii'} T_{jj'} (C_{ij}^{\text{s}} - C_{i'j'}^{\text{t}}) \le 0, \\ -\epsilon, & \text{otherwise.} \end{cases}$$
>
> *solves the inner maximization problem*
>
> $$\max_{\mathbf{E} \in \mathcal{U}_\epsilon} f(\mathbf{T}, \mathbf{E}).$$

2. *RGWD is lower bounded, that is,*

$$\mathrm{RGWD}\left((\mathbf{C}^{\mathrm{s}}, \mathbf{p}^{\mathrm{s}}), (\mathbf{C}^{\mathrm{t}}, \mathbf{p}^{\mathrm{t}}), \epsilon\right) \geq \epsilon,$$

*where the equality holds if and only if there exists a bijective $\pi^* : [\![n^{\mathrm{s}}]\!] \to [\![n^{\mathrm{t}}]\!]$ such that $p_i^{\mathrm{s}} = p_{\pi^*(i)}^{\mathrm{t}}$ for all $i \in [\![n^{\mathrm{s}}]\!]$ and $C_{ij}^{\mathrm{s}} = C_{\pi^*(i)\pi^*(j)}^{\mathrm{t}}$ for all $i, j \in [\![n^{\mathrm{s}}]\!]$.*

3. *The triangle inequality holds for RGWD, i.e.,*

$$\mathrm{RGWD}\left((\mathbf{C}^1, \mathbf{p}^1), (\mathbf{C}^3, \mathbf{p}^3), \epsilon\right)$$
$$\leq \mathrm{RGWD}\left((\mathbf{C}^1, \mathbf{p}^1), (\mathbf{C}^2, \mathbf{p}^2), \epsilon\right) + \mathrm{RGWD}\left((\mathbf{C}^2, \mathbf{p}^2), (\mathbf{C}^3, \mathbf{p}^3), \epsilon\right).$$

4. *RGWD is invariant to the permutation of node orders, i.e., for all permutation matrices $\mathbf{Q}^{\mathrm{s}}$ and $\mathbf{Q}^{\mathrm{t}}$,*

$$\mathrm{RGWD}\left((\mathbf{C}^{\mathrm{s}}, \mathbf{p}^{\mathrm{s}}), (\mathbf{C}^{\mathrm{t}}, \mathbf{p}^{\mathrm{t}}), \epsilon\right) = \mathrm{RGWD}\left((\mathbf{Q}^{\mathrm{s}\top}\mathbf{C}^{\mathrm{s}}\mathbf{Q}^{\mathrm{s}}, \mathbf{Q}^{\mathrm{s}\top}\mathbf{p}^{\mathrm{s}}), (\mathbf{Q}^{\mathrm{t}\top}\mathbf{C}^{\mathrm{t}}\mathbf{Q}^{\mathrm{t}}, \mathbf{Q}^{\mathrm{t}\top}\mathbf{p}^{\mathrm{t}}), \epsilon\right).$$

As is implied by Theorem 1, RGWD does not define a distance between metric-measure spaces. Firstly, the identity axiom is not satisfied. Secondly, the symmetry generally does not hold either, which we exemplify below.

**Example 1 (Asymmetry of RGWD)** *Consider the case $\mathbf{p}^{\mathrm{s}} = \mathbf{p}^{\mathrm{t}} = \left[\begin{smallmatrix} 0.5 \\ 0.5 \end{smallmatrix}\right]$, $\mathbf{C}^{\mathrm{s}} = \left[\begin{smallmatrix} 0 & 1 \\ 1 & 0 \end{smallmatrix}\right]$, $\mathbf{C}^{\mathrm{t}} = \left[\begin{smallmatrix} 0 & 4 \\ 4 & 0 \end{smallmatrix}\right]$. Then $\mathrm{RGWD}\left((\mathbf{C}^{\mathrm{s}}, \mathbf{p}^{\mathrm{s}}), (\mathbf{C}^{\mathrm{t}}, \mathbf{p}^{\mathrm{t}}), 1\right) = 11.5$ with the solution given by $\mathbf{T}^* = \left[\begin{smallmatrix} 0.25 & 0.25 \\ 0.25 & 0.25 \end{smallmatrix}\right]$ and $\mathbf{E}^* = \left[\begin{smallmatrix} -1 & 1 \\ 1 & -1 \end{smallmatrix}\right]$. In contrast, $\mathrm{RGWD}\left((\mathbf{C}^{\mathrm{t}}, \mathbf{p}^{\mathrm{t}}), (\mathbf{C}^{\mathrm{s}}, \mathbf{p}^{\mathrm{s}}), 1\right) = 10.5$ with $\mathbf{T}^* = \left[\begin{smallmatrix} 0.25 & 0.25 \\ 0.25 & 0.25 \end{smallmatrix}\right]$ and $\mathbf{E}^* = \left[\begin{smallmatrix} -1 & -1 \\ -1 & -1 \end{smallmatrix}\right]$. One has $\mathrm{RGWD}\left((\mathbf{C}^{\mathrm{s}}, \mathbf{p}^{\mathrm{s}}), (\mathbf{C}^{\mathrm{t}}, \mathbf{p}^{\mathrm{t}}), 1\right) \neq \mathrm{RGWD}\left((\mathbf{C}^{\mathrm{t}}, \mathbf{p}^{\mathrm{t}}), (\mathbf{C}^{\mathrm{s}}, \mathbf{p}^{\mathrm{s}}), 1\right).$*

Example 1 showcases that RGWD is asymmetric even if $n^{\mathrm{s}} = n^{\mathrm{t}}$.

### 3.2 Numerical Scheme of RGWD

We derive a gradient based numerical scheme to solve RGWD by exploiting the property that the inner maximization problem has a closed-form solution, which is summarized in Algorithm 1. In each iteration, $\mathbf{E}_\tau$ that solves the inner problem for current $\mathbf{T}_\tau$ is calculated. Then, the transport plan is updated using the projected gradient descent.

---

**Algorithm 1** Projected Gradient Descent for RGWD

---

1: **Input:** Initialization $\mathbf{T}_0$, step-size $\eta$, number of iterations $N$.
2: **Output:** Estimated optimal transport plan $\hat{\mathbf{T}}$ and its corresponding perturbation $\hat{\mathbf{E}}$.
3: **for** $\tau = 0, 1, \ldots, N - 1$ **do**
4:     Find $\mathbf{E}_\tau$ that maximizes $f(\mathbf{T}_\tau, \mathbf{E})$.
5:     Update the transport plan via

$$\mathbf{T}_{\tau+1} = \mathrm{Proj}_{\Pi(\mathbf{p}^{\mathrm{s}}, \mathbf{p}^{\mathrm{t}})} \left(\mathbf{T}_\tau - \eta \nabla_{\mathbf{T}} f(\mathbf{T}_\tau, \mathbf{E}_\tau)\right),$$

    where the partial gradient takes the form

$$\nabla_{\mathbf{T}} f(\mathbf{T}_\tau, \mathbf{E}_\tau) = 2(\mathbf{C}^{\mathrm{s}} \odot \mathbf{C}^{\mathrm{s}}) \mathbf{T}_\tau \mathbf{1}^{n^{\mathrm{t}}} \mathbf{1}^{n^{\mathrm{t}}\top} 2\mathbf{1}^{n^{\mathrm{s}}} \mathbf{1}^{n^{\mathrm{s}}\top} \mathbf{T}_\tau \left((\mathbf{C}^{\mathrm{t}} + \mathbf{E}_\tau) \odot (\mathbf{C}^{\mathrm{t}} + \mathbf{E}_\tau)\right)$$
$$-4\mathbf{C}^{\mathrm{s}} \mathbf{T}_\tau (\mathbf{C}^{\mathrm{t}} + \mathbf{E}_\tau),$$

    with $\odot$ denoting the element-wise multiplication.
6: **end for**
7: Pick $\tau$ uniformly at random from $\{1, 2, \ldots, T\}$.
8: Set $\hat{\mathbf{T}} \leftarrow \mathbf{T}_\tau$.
9: Find $\hat{\mathbf{E}}$ that maximizes $f(\hat{\mathbf{T}}, \mathbf{E})$.

---

To present the convergence guarantee of Algorithm 1, we introduce the notion of the Moreau envelope. The stationarity of any function $h(\mathbf{x})$ can be quantified by the norm of the gradient of its Moreau envelope $h_\lambda(\mathbf{x}) = \min_{\mathbf{x}'} h(\mathbf{x}') + \frac{1}{2\lambda}\|\mathbf{x} - \mathbf{x}'\|^2$. The following theorem gives the convergence rate of Algorithm 1 and the proof is deferred to the appendix.

**Theorem 2** *Define* $\phi(\cdot) = \max_{\mathbf{E} \in \mathcal{U}_\epsilon} f(\cdot, \mathbf{E})$. *The output* $\hat{\mathbf{T}}$ *of Algorithm 1 with step-size* $\eta = \frac{\gamma}{\sqrt{N+1}}$ *satisfies*

$$\mathbb{E}\big[\|\nabla\phi_{1/2l}(\hat{\mathbf{T}})\|^2\big] \leq 2\frac{\phi_{1/2l}(\mathbf{T}_0) - \min_{\mathbf{T} \in \Pi(\mathbf{p}^s, \mathbf{p}^t)} \phi(\mathbf{T}) + lL^2\gamma^2}{\gamma\sqrt{N+1}},$$

*where* $l = \sqrt{2}\max\{10n^3U_1^2 + 6n^3U_1\epsilon + 4nU_1U_2 + 4n^3\epsilon^2, 6n^2U_1U_2 + 2U_2^2 + 4n^2U_2\epsilon\}$ *and* $L = \sqrt{2}\max\{(4U_1+2\epsilon)U_2^2n^3, 2(2U_1+\epsilon)^2U_2n^3\}$ *with* $n = \max\{n^s, n^t\}$, $U_1 = \max\{\|\mathbf{C}^s\|_\infty, \|\mathbf{C}^t\|_\infty\}$ *and* $U_2 = \max\{\|\mathbf{p}^t\|_2, \max_{\mathbf{T}' \in \Pi(\mathbf{p}^s, \mathbf{p}^t)} \|\mathbf{T}'\|_F\}$.

When $U_1$ and $\epsilon$ are of the order $\mathcal{O}(\frac{1}{n^3})$, both $l$ and $L$ are of the order $\mathcal{O}(1)$ and Theorem 2 states that an $\delta$-stationary solution can be obtained within $\mathcal{O}(\frac{1}{\delta^2})$ iterations. Note that we can multiply $\mathbf{C}^s$, $\mathbf{C}^t$, and $\epsilon$ by the same number without affecting the resulted transport plan.

## 4 ROBUST GRAPH DICTIONARY LEARNING

The problem of learning a robust dictionary for graph data is now formulated as follows. Given a dataset of $K$ graphs expressed by $\{(\mathbf{C}^k, \mathbf{p}^k)\}_{k \in [\![K]\!]}$, estimating the optimal dictionary is formalized by

$$\min_{\{\bar{\mathbf{C}}^m\}_{m \in [\![M]\!]}, \{\mathbf{w}^k\}_{k \in [\![K]\!]}} \sum_{k=1}^{K} \text{RGWD}^2\left(\left(\sum_{m=1}^{M} w_m^k \bar{\mathbf{C}}^m, \bar{\mathbf{p}}\right), (\mathbf{C}^k, \mathbf{p}^k), \epsilon\right) - \lambda\|\mathbf{w}^k\|^2, \qquad (2)$$

where $\{\bar{\mathbf{C}}^m\}_{m \in [\![M]\!]}$ and $\{\mathbf{w}^k\}_{k \in [\![K]\!]}$ are the dictionary and graph embeddings respectively, and $\bar{\mathbf{p}}$ is obtained by sorting and averaging $\{\mathbf{p}^k\}_{k \in [\![K]\!]}$ following Xu et al. (2019a). To resolve (2), we propose a nested iterative optimization algorithm that is summarized in Algorithm 2. The main idea is that the dictionary and embeddings are updated alternatingly. We discuss some crucial details below.

---

**Algorithm 2** Robust Graph Dictionary Learning (RGDL)

1: **Input:** The dataset $\{\mathbf{C}^k, \mathbf{p}^k\}_{k \in [\![K]\!]}$, the initial dictionary $\{\bar{\mathbf{C}}^m\}_{m \in [\![M]\!]}$, the number of iterations $T$, mini-batch size $b$.
2: **Output:** The learned dictionary $\{\bar{\mathbf{C}}^m\}_{m \in [\![M]\!]}$.
3: **for** $t = 0, 1, \ldots, T-1$ **do**
4:     Sample a mini-batch of graphs whose indices are denoted by $\mathcal{B}$ such that $|\mathcal{B}| = b$.
5:     **for** $k \in \mathcal{B}$ **do**
6:         Initialize $\mathbf{w}^k = \frac{1}{M}\mathbf{1}^M$ and $\mathbf{T}^k = \bar{\mathbf{p}}\mathbf{p}^{k^\top}$.
7:         **repeat**
8:             Calculate $(\mathbf{T}^k, \mathbf{E}^k)$ via Algorithm 1 with fixed $\mathbf{w}^k$.
9:             Compute $\mathbf{w}^k$ solving (4) for the fixed $\mathbf{T}^k$ and $\mathbf{E}^k$ with conditional gradient.
10:         **until** Convergence
11:     **end for**
12:     Update the atom $\bar{\mathbf{C}}^m$ for $m \in [\![M]\!]$ with stochastic gradient $\hat{\nabla}_{\bar{\mathbf{C}}^m}$ which has the form

$$\hat{\nabla}_{\bar{\mathbf{C}}^m} = \frac{2}{b}\sum_{k \in \mathcal{B}} w_m^k\left(\sum_{m'=1}^{M} w_{m'}^k \bar{\mathbf{C}}^{m'} \odot \bar{\mathbf{p}}\bar{\mathbf{p}}^\top - \mathbf{T}^k(\mathbf{C}^k + \mathbf{E}^k)\mathbf{T}^{k^\top}\right). \qquad (3)$$

13: **end for**

---

**Solving $\mathbf{w}^k$.** We now formulate the problem of obtaining the embedding of the $k^{\text{th}}$ graph $\mathcal{G}^k$ when the dictionary is fixed and the PRM is inaccurate. Given dictionary $\{\bar{\mathbf{C}}^m\}_{m \in [\![M]\!]}$ where each $\bar{\mathbf{C}}^m \in \mathbb{R}^{n^a \times n^a}$, the embedding of $\mathcal{G}^k$ expressed by $(\mathbf{C}^k, \mathbf{p}^k)$ is calculated by solving

$$\min_{\mathbf{w}^k \in \Delta^M} \text{RGWD}^2\left(\left(\sum_{m=1}^{M} w_m^k \bar{\mathbf{C}}^m, \bar{\mathbf{p}}\right), (\mathbf{C}^k, \mathbf{p}^k), \epsilon\right) - \lambda\|\mathbf{w}^k\|^2, \qquad (4)$$

where $\lambda \geq 0$ induces a negative quadratic regularization promoting sparsity on the simplex (Li et al., 2020; Vincent-Cuaz et al., 2021). When $\mathbf{w}^k$ is fixed, updating $\mathbf{T}^k$ and $\mathbf{E}^k$ can be solved by Algorithm 1 whose convergence is guaranteed by Theorem 2. For fixed $\mathbf{T}^k$ and $\mathbf{E}^k$, the problem of updating $\mathbf{w}^k$ is a non-convex problem that can be tackled by a conditional gradient algorithm. Note that for non-convex problems, the conditional gradient algorithm is proved to converge to a local stationary point (Lacoste-Julien, 2016). Such a procedure is described from Line 5 to Line 11 in Algorithm 2, which we observe converges within tens of iterations empirically.

**Stochastic updates.** To enhance computational efficiency, each atom is updated with stochastic estimates of the gradient. At each stochastic update, $b$ embedding learning problems are solved independently for the current dictionary using the procedure stated above, where $b$ is the size of the sampled mini-batch. Each atom is then updated using the stochastic gradient given in Eq. (3). Note that the symmetry of each atom is preserved as long as the initialized atom is symmetric, since the stochastic gradients are symmetric.

## 5 EXPERIMENTS

This section provides empirical evidence that RGDL performs well in the unsupervised graph clustering task on both synthetic and real-world datasets[1]. The heat kernel matrix is employed for the PRM since it captures both global and local topology and achieves good performance in many tasks (Donnat et al., 2018; Tsitsulin et al., 2018; Chowdhury and Needham, 2021).

### 5.1 SIMULATED DATASETS

We first test RGDL in the graph clustering task on datasets simulated according to the well-studied Stochastic Block Model (SBM) (Holland et al., 1983; Wang and Wong, 1987). RGDL is compared against the following state-of-the-art graph clustering methods: (i) GDL (Vincent-Cuaz et al., 2021) learns graph dictionaries via GWD; (ii) Gromov–Wasserstein Factorization (GWF) (Xu, 2020) that approximates graphs via GW barycenters; (iii) Spectral Clustering (SC) of Shi and Malik (2000); Stella and Shi (2003) applied to the matrix with each entry storing the GWD between two graphs.

Table 1: Average (stdev) ARI scores for the first scenario of synthetic datasets.

| | balanced | | | unbalanced | | |
|---|---|---|---|---|---|---|
| $\sigma$ | 0.05 | 0.10 | 0.15 | 0.05 | 0.10 | 0.15 |
| **GDL** | 0.119(0.017) | 0.031(0.012) | 0.016(0.006) | 0.049(0.019) | 0.018(0.004) | 0.018(0.001) |
| **GWF** | 0.071(0.007) | 0.034(0.003) | 0.008(0.001) | 0.052(0.020) | 0.014(0.001) | 0.015(0.001) |
| **SC** | 0.057(0.002) | 0.033(0.002) | 0.010(0.001) | 0.054(0.024) | 0.015(0.004) | 0.010(0.001) |
| **RGDL($\epsilon$=0.01)** | 0.316(0.005) | 0.161(0.005) | 0.052(0.002) | 0.246(0.013) | 0.039(0.009) | 0.024 (0.001) |
| **RGDL($\epsilon$=0.1)** | 0.853(0.003) | 0.756(0.018) | 0.439(0.015) | 0.765(0.022) | 0.694(0.046) | 0.499(0.016) |
| **RGDL($\epsilon$=0.2)** | **0.975(0.025)** | 0.879(0.023) | 0.736(0.020) | 0.866(0.023) | 0.815(0.028) | 0.770(0.016) |
| **RGDL($\epsilon$=0.3)** | **0.975(0.025)** | 0.879(0.023) | 0.869(0.013) | **0.943(0.001)** | 0.916(0.027) | 0.848(0.061) |
| **RGDL($\epsilon$=10)** | **0.975(0.025)** | **0.950(0.000)** | **0.950(0.000)** | **0.943(0.001)** | **0.943(0.001)** | **0.943(0.001)** |
| **RGDL($\epsilon$=30)** | 0.781(0.046) | 0.779(0.070) | 0.728(0.085) | 0.723(0.067) | 0.698(0.057) | 0.666(0.040) |

**Dataset generation.** We consider two scenarios of inaccuracies. In the first scenario (S1), Gaussian noise is added into the heat kernel matrix of each graph. More specifically, denoting the heat kernel matrix of the $k^{\text{th}}$ graph as $\mathbf{C}^{k*}$ for $k \in [\![K]\!]$, the PRM available to DL methods is $\mathbf{C}^k = \mathbf{C}^{k*} + \mathbf{Z} + \mathbf{Z}^\top$ where each entry $Z_{ij}$ of $\mathbf{Z}$ is sampled from the Gaussian distribution $\mathcal{N}(0, \sigma)$. In the second scenario (S2), we randomly add $\rho|\mathcal{E}|$ edges into the graph and then randomly remove $\rho|\mathcal{E}|$ edges while keeping the graph connected, where $\mathcal{E}$ is the edge set of the graph. The heat kernel matrix is then constructed for the modified graph. Such two scenarios allow us to study the performance of RGDL against different scales of inaccuracies. In both S1 and S2, we generate two datasets, both of which involve three generative structures (also used to label graphs): dense (only one community), two communities, and three communities. We fix $p = 0.1$ as the probability of inter-community connectivity and $1 - p$ as the probability of intra-community connectivity. The first dataset includes 20 graphs for each generative structure and thus is referred to as the balanced dataset. The second

---

[1]Code available at `https://github.com/cxxszz/rgdl`.

Table 2: Average (stdev) ARI scores for the second scenario of synthetic datasets.

| $\rho$ | balanced | | | unbalanced | | |
|---|---|---|---|---|---|---|
| | 0.00 | 0.05 | 0.10 | 0.00 | 0.05 | 0.1 |
| **GDL** | 0.260(0.020) | 0.187(0.013) | 0.024(0.004) | 0.152(0.008) | 0.018(0.001) | 0.005(0.001) |
| **GWF** | 0.182(0.006) | 0.086(0.004) | 0.027(0.005) | 0.020(0.005) | 0.016(0.002) | 0.010(0.002) |
| **SC** | 0.204(0.002) | 0.129(0.017) | 0.016(0.005) | 0.129(0.008) | 0.013(0.001) | 0.011(0.005) |
| **RGDL**($\epsilon$=0.01) | 0.451(0.014) | 0.449(0.016) | 0.449(0.016) | 0.401(0.002) | 0.401(0.002) | 0.399(0.000) |
| **RGDL**($\epsilon$=0.1) | **1.000(0.000)** | **1.000(0.000)** | 0.975(0.025) | **1.000(0.000)** | **1.000(0.000)** | **1.000(0.000)** |
| **RGDL**($\epsilon$=0.2) | **1.000(0.000)** | **1.000(0.000)** | 0.975(0.025) | **1.000(0.000)** | **1.000(0.000)** | **1.000(0.000)** |
| **RGDL**($\epsilon$=0.3) | **1.000(0.000)** | **1.000(0.000)** | **1.000(0.000)** | **1.000(0.000)** | **1.000(0.000)** | **1.000(0.000)** |
| **RGDL**($\epsilon$=10) | **1.000(0.000)** | **1.000(0.000)** | **1.000(0.000)** | **1.000(0.000)** | **1.000(0.000)** | **1.000(0.000)** |
| **RGDL**($\epsilon$=30) | 0.896(0.070) | 0.888(0.080) | 0.857(0.044) | 0.864(0.057) | 0.827(0.043) | 0.816(0.074) |

dataset consists of 12, 18, and 30 graphs for the three generative structures respectively, and is hence named as the unbalanced dataset. The number of graph nodes is uniformly sampled from $[30, 50]$. The magnitude of the observed PRM $\mathbf{C}^k$ satisfies $\|\mathbf{C}^k\|_\infty \leq 15$.

**Evaluating the performance.** The learned embeddings of the graphs are used as input for a k-means algorithm to cluster graphs. We use the well-known Adjusted Rand Index (ARI) (Hubert and Arabie, 1985; Steinley, 2004), to evaluate the quality of clustering by comparing it with the graph labels. RGDL with varied $\epsilon$ is compared against GDL, GWF, and SC. RGDL, GDL and GWF use three atoms which are $\mathbb{R}^{6\times6}$ matrices. We run each method for 5 times and report the averaged ARI scores and the standard deviations. Experimental results reported in Table 1 and Table 2 demonstrate RGDL outperforms baselines significantly.

**Influence of $\epsilon$.** RGDL with moderate $\epsilon$ values outperforms baseline methods by a large margin and is more robust to the noise. Even when $\epsilon$ is relatively small ($\epsilon$=0.01), RGDL achieves better performance than baselines. Increasing $\epsilon$ within a suitable range can boost ARI and RGDL is not sensitive to the choice of $\epsilon$. If $\epsilon$ becomes too large, the performance of RGDL slowly decreases. In practice, when a small quantity of data labels are available, $\epsilon$ can be chosen according to the performance on this small subset of data.

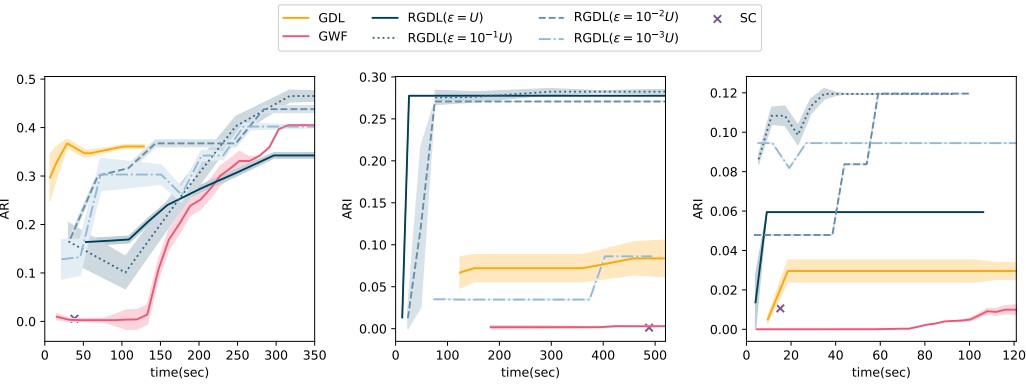

Figure 1: ARI scores vs. time on MUTAG (left), BZR(middle), and Peking_1(right) datasets.

## 5.2 REAL-WORLD DATASETS

We further use RGDL to cluster real-world graphs. We consider widely utilized benchmark datasets including MUTAG (Debnath et al., 1991), BZR (Sutherland et al., 2003), and Peking_1 (Pan et al., 2016). The labels of the graphs are employed as the ground truth to evaluate the estimated clustering results. For each dataset, the size of the atoms is set as the median of the numbers of graph nodes following Vincent-Cuaz et al. (2021). The number of atoms $M$ is set as $M = \beta(\#\text{classes})$ where $\beta$ is chosen from $\{2, 3, 4, 5\}$. RGDL is run with different values of $\epsilon$. Specifically, $\epsilon$ is chosen from $\{U, 10^{-1}U, 10^{-2}U, 10^{-3}U\}$ where $U = \max_{k\in[\![K]\!]} \|\mathbf{C}_k\|_\infty$.

**Results.** The experimental results on real-world graphs are reported in Figure 1. RGDL with $\epsilon = 10^{-1}U$ or $\epsilon = 10^{-2}U$ outperforms baselines on all datasets, which implies that the observed graphs contain structural noise and $\left[10^{-2}U, 10^{-1}U\right]$ is often a suitable range for $\epsilon$. The time required for RGDL to converge is comparable to that of state-of-the-art of methods.

## 6 RELATED WORK

**Unbalanced OT.** Enhancing the robustness of the optimal transport plan has received wide attention recently (Balaji et al., 2020; Mukherjee et al., 2021; Le et al., 2021; Nietert et al., 2022; Chapel et al., 2020; Séjourné et al., 2021; Vincent-Cuaz et al., 2022). Originally, robust variants of classical OT were proposed to compare distributions supported on the *same* metric space (Balaji et al., 2020; Mukherjee et al., 2021; Le et al., 2021; Nietert et al., 2022), which model the noise as outlier supports and reduce the influence of outlier supports by allowing mass destruction and creation. Following the same spirit, variants of the GW distance that also relax the marginal constraints were proposed (Chapel et al., 2020; Séjourné et al., 2021; Vincent-Cuaz et al., 2022). However, these methods do not take the inaccuracies of the pairwise distances/similarities into account. The proposed RGWD aims to handle such cases.

**Graph representation learning and graph comparison.** Comparing graphs often requires learning meaningful graph representations. Some methods manually design representations that are invariant under graph isomorphism (Bagrow and Bollt, 2019; Tam and Dunson, 2022). Such representations are often sophisticated and require domain knowledge. Graph neural network-based methods learn the representations of graphs in an end-to-end manner (Scarselli et al., 2008; Zhang et al., 2018; Lee et al., 2018; Errica et al., 2019), which however requires a large amount of labeled data. Another family of methods that uses graph representations implicitly is referred to as graph kernels (Shervashidze et al., 2009; Vishwanathan et al., 2010). GWD and its variants based methods can estimate the node correspondence and provide an interpretable discrepancy between compared graphs (Xu et al., 2019b; Titouan et al., 2019; Barbe et al., 2020; Chapel et al., 2020). In this paper, we propose a novel graph dictionary learning method based on a robust variant of GWD to learn representations of graphs which are useful in downstream tasks.

**Non-linear combination of atoms.** Classic DL methods are linear in the sense that they attempt to approximate each vectorized datum by a linear combination of a few basis elements. Recently, non-linear operations were also considered. In order to exploit the non-linear nature of data, Autoencoder-based methods encode them to low-dimensional vectors using neural networks, and decode data with another neural network (Hinton and Salakhutdinov, 2006; Hu and Tan, 2018). Another family of methods replace the linear combinations by geodesic interpolations (Boissard et al., 2011; Bigot et al., 2013; Seguy and Cuturi, 2015; Schmitz et al., 2018). More closely related to our work, Xu 2020 proposed to approximate graphs via the GW barycenter of graph atoms, which however involves a complicated and computational demanding optimization problem.

**Projection robust OT.** To improve the convergence of empirical Wasserstein distances (Rüschendorf, 1985) to population ones, a group of methods project the distributions to informative low-dimensional subspaces (Paty and Cuturi, 2019; Lin et al., 2020; 2021), which involves solving min-max or max-min problems. This paper considers distributions supported on different metric spaces and does not project distributions.

## 7 CONCLUSION

In this paper, we propose a novel graph dictionary learning algorithm that is robust to the structural noise of graphs. We first propose a robust variant of GWD, referred to as RGWD, which involves a minimax optimization problem. Exploiting the fact that the inner maximization problem has a closed-form solution, an efficient numerical scheme is derived. Based on RGWD, a robust dictionary learning algorithm for graphs called RGDL is derived to learn atoms from noisy graph data. Numerical results on both simulated and real-world datasets demonstrate that RGDL achieves good performance in the presence of structural noise.

## ACKNOWLEDGMENTS

This work is supported by the National Key R&D Program of China (2020YFB1313501, 2020AAA0107400), National Natural Science Foundation of China (T2293723, 61972347, 61672376, 61751209, 61472347, 62206248), Zhejiang Provincial Natural Science Foundation (LR19F020005, LZ18F020002), the Key Research and Development Project of Zhejiang Province (2022C01022, 2022C01119, 2021C03003), the Fundamental Research Funds for the Central Universities (226-2022-00051), Alibaba-Zhejiang University Joint Research Institute of Frontier Technologies, China Scholarship Council (202206320311), and MEXT KAKENHI (20H04243).

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

# Appendix

The appendix is organized as follows. We first provide omitted proofs in the main paper in Sec. A. Then, algorithmic details are presented in Sec. B. Finally, Sec. C gives additional experimental results.

## A OMITTED PROOFS

**Theorem 1** *Given an observed source graph $\mathcal{G}^s$ and a target graph $\mathcal{G}^t$ that can be expressed as $(\mathbf{C}^s, \mathbf{p}^s)$ and $(\mathbf{C}^t, \mathbf{p}^t)$ respectively, RGWD satisfies*

1. *for all $\mathbf{T} \in \Pi(\mathbf{p}^s, \mathbf{p}^t)$, $\mathbf{E}(\mathbf{T}) = [E_{i'j'}(\mathbf{T})]$ where*

$$E_{i'j'}(\mathbf{T}) = \begin{cases} \epsilon, & \text{if } \sum_{i,j=1}^{n^s} T_{ii'}T_{jj'}(C_{ij}^s - C_{i'j'}^t) \leq 0, \\ -\epsilon, & \text{otherwise.} \end{cases}$$

*solves the inner maximization problem*

$$\max_{\mathbf{E} \in \mathcal{U}_\epsilon} f(\mathbf{T}, \mathbf{E}).$$

2. *RGWD is lower bounded, that is,*

$$\text{RGWD}\big((\mathbf{C}^s, \mathbf{p}^s), (\mathbf{C}^t, \mathbf{p}^t), \epsilon\big) \geq \epsilon,$$

*where the equality holds if and only if there exists a bijective $\pi^* : [\![n^s]\!] \to [\![n^t]\!]$ such that $p_i^s = p_{\pi^*(i)}^t$ for all $i \in [\![n^s]\!]$ and $C_{ij}^s = C_{\pi^*(i)\pi^*(j)}^t$ for all $i, j \in [\![n^s]\!]$.*

3. *The triangle inequality holds for RGWD, i.e.,*

$$\begin{aligned} &\text{RGWD}\big((\mathbf{C}^1, \mathbf{p}^1), (\mathbf{C}^3, \mathbf{p}^3), \epsilon\big) \\ &\leq \text{RGWD}\big((\mathbf{C}^1, \mathbf{p}^1), (\mathbf{C}^2, \mathbf{p}^2), \epsilon\big) + \text{RGWD}\big((\mathbf{C}^2, \mathbf{p}^2), (\mathbf{C}^3, \mathbf{p}^3), \epsilon\big). \end{aligned}$$

4. *RGWD is invariant to the permutation of node orders, i.e., for all permutation matrices $\mathbf{Q}^s$ and $\mathbf{Q}^t$,*

$$\text{RGWD}\big((\mathbf{C}^s, \mathbf{p}^s), (\mathbf{C}^t, \mathbf{p}^t), \epsilon\big) = \text{RGWD}\big((\mathbf{Q}^{s\top}\mathbf{C}^s\mathbf{Q}^s, \mathbf{Q}^{s\top}\mathbf{p}^s), (\mathbf{Q}^{t\top}\mathbf{C}^t\mathbf{Q}^t, \mathbf{Q}^{t\top}\mathbf{p}^t), \epsilon\big).$$

**Proof:** (i) The objective can be rewritten as follows,

$$\sum_{i,j=1}^{n^s} \sum_{i',j'=1}^{n^t} (C_{ij}^s - C_{i'j'}^t - E_{i'j'})^2 T_{ii'}T_{jj'}$$

$$= \sum_{i,j=1}^{n^s} \sum_{i',j'=1}^{n^t} (C_{ij}^s - C_{i'j'}^t)^2 T_{ii'}T_{jj'} + \sum_{i',j'=1}^{n^t} \left( \sum_{i,j=1}^{n^s} T_{ii'}T_{jj'}E_{i'j'}^2 - 2\sum_{i,j=1}^{n^s} T_{ii'}T_{jj'}(C_{ij}^s - C_{i'j'}^t)E_{i'j'} \right)$$

$$= \sum_{i,j=1}^{n^s} \sum_{i',j'=1}^{n^t} (C_{ij}^s - C_{i'j'}^t)^2 T_{ii'}T_{jj'} + \sum_{i',j'=1}^{n^t} \left( p_{i'}^t p_{j'}^t E_{i'j'}^2 - 2\sum_{i,j=1}^{n^s} T_{ii'}T_{jj'}(C_{ij}^s - C_{i'j'}^t)E_{i'j'} \right), \tag{5}$$

which by the property of quadratic functions yields the closed-form solution $\mathbf{E}(\mathbf{T}) = [E_{i'j'}(\mathbf{T})]$, where

$$E_{i'j'}(\mathbf{T}) = \begin{cases} \epsilon, & \text{if } \sum_{i,j=1}^{n^s} T_{ii'}T_{jj'}(C_{ij}^s - C_{i'j'}^t) \leq 0, \\ -\epsilon, & \text{otherwise.} \end{cases}$$

It is easy to verify that the such a choice guarantees the symmetry of $\mathbf{E}(\mathbf{T})$.

(ii) We now prove the lower boundedness. By Eq. (5), we have

$$\min_{\mathbf{T} \in \Pi(\mathbf{p}^s, \mathbf{p}^t)} \max_{\mathbf{E} \in \mathcal{U}_\epsilon} \sum_{i,j=1}^{n^s} \sum_{i',j'=1}^{n^t} (C_{ij}^s - C_{i'j'}^t - E_{i'j'})^2 T_{ii'}T_{jj'} \geq \min_{\mathbf{T} \in \Pi(\mathbf{p}^s, \mathbf{p}^t)} \sum_{i,j=1}^{n^s} \sum_{i',j'=1}^{n^t} T_{ii'}T_{jj'}\epsilon^2 = \epsilon^2.$$

Note that when there exists a bijective $\pi^* : [\![n^s]\!] \to [\![n^t]\!]$ such that $p_i^s = p_{\pi^*(i)}^t$ for all $i \in [\![n^s]\!]$ and $C_{ij}^s = C_{\pi^*(i)\pi^*(j)}^t$ for all $i, j \in [\![n^s]\!]$, choosing the transport plan $\hat{\mathbf{T}} = [\hat{T}_{ii'}]$ where

$$
\hat{T}_{ii'} = \begin{cases} p_i^s, & \text{if } i' = \pi^*(i), \\ 0, & \text{otherwise,} \end{cases}
$$

we have for all $i', j' \in [\![n^t]\!]$,

$$
\sum_{i,j=1}^{n^s} \hat{T}_{ii'} \hat{T}_{jj'} (C_{ij}^s - C_{i'j'}^t) = \hat{T}_{\pi^{*-1}(i')i'} \hat{T}_{\pi^{*-1}(j')j'} \left( C_{\pi^{*-1}(i')\pi^{*-1}(j')}^s - C_{i'j'}^t \right) = 0,
$$

which implies that $E_{i'j'}(\mathbf{T}) = \epsilon$ for all $i', j' \in [\![n^t]\!]$. We then have

$$
\sum_{i,j=1}^{n^s} \sum_{i',j'=1}^{n^t} (C_{ij}^s - C_{i'j'}^t - E_{i'j'})^2 \hat{T}_{ii'} \hat{T}_{jj'} = \sum_{i,j=1}^{n^s} (C_{ij}^s - C_{\pi^*(i)\pi^*(j)}^t - \epsilon)^2 \hat{T}_{i\pi^*(i)} \hat{T}_{j\pi^*(j)} = \epsilon^2.
$$

Therefore, in such a case, $\mathrm{RGWD}\left((\mathbf{C}^s, \mathbf{p}^s), (\mathbf{C}^t, \mathbf{p}^t), \epsilon\right) = \epsilon$. On the other hand, when such a bijective does not exist,

$$
\mathrm{RGWD}\left((\mathbf{C}^s, \mathbf{p}^s), (\mathbf{C}^t, \mathbf{p}^t), \epsilon\right) \geq \sqrt{\epsilon^2 + \min_{\mathbf{T} \in \Pi(\mathbf{p}^s, \mathbf{p}^t)} \sum_{i,j=1}^{n^s} \sum_{i',j'=1}^{n^t} (C_{ij}^s - C_{i'j'}^t)^2 T_{ii'} T_{jj'}} > \epsilon,
$$

where the strict inequality is due to the fact that $\sum_{i,j=1}^{n^s} \sum_{i',j'=1}^{n^t} (C_{ij}^s - C_{i'j'}^t)^2 T_{ii'} T_{jj'} > 0$.
(iii) Thirdly, we prove the triangle inequality. Given tuples $(\mathbf{C}^1, \mathbf{p}^1)$, $(\mathbf{C}^2, \mathbf{p}^2)$, and $(\mathbf{C}^3, \mathbf{p}^3)$ which have the node numbers $n^1$, $n^2$, and $n^3$ respectively, let $(\mathbf{T}^{*12}, \mathbf{E}^{*12})$, $(\mathbf{T}^{*23}, \mathbf{E}^{*23})$, and $(\mathbf{T}^{*13}, \mathbf{E}^{*13})$ be the solutions of $\mathrm{RGWD}\left((\mathbf{C}^1, \mathbf{p}^1), (\mathbf{C}^2, \mathbf{p}^2), \epsilon\right)$, $\mathrm{RGWD}\left((\mathbf{C}^2, \mathbf{p}^2), (\mathbf{C}^3, \mathbf{p}^3), \epsilon\right)$, and $\mathrm{RGWD}\left((\mathbf{C}^1, \mathbf{p}^1), (\mathbf{C}^3, \mathbf{p}^3), \epsilon\right)$. Define $\mathbf{T}^{13} = [T_{i_1 i_3}^{13}]$ where

$$
T_{i_1 i_3}^{13} = \sum_{i_2=1}^{n^2} \frac{T_{i_1 i_2}^{*12} T_{i_2 i_3}^{*23}}{p_{i_2}^2}.
$$

Then we have
$\mathrm{RGWD}\left((\mathbf{C}^1, \mathbf{p}^1), (\mathbf{C}^3, \mathbf{p}^3), \epsilon\right)$

$$
\leq \sqrt{\sum_{i_1,j_1=1}^{n^1} \sum_{i_3,j_3=1}^{n^3} \left(C_{i_1 j_1}^1 - C_{i_3 j_3}^3 - E_{i_3 j_3}^{*13}\right)^2 T_{i_1 i_3}^{13} T_{j_1 j_3}^{13}}
$$

$$
= \sqrt{\sum_{i_1,j_1=1}^{n^1} \sum_{i_3,j_3=1}^{n^3} \left(C_{i_1 j_1}^1 - C_{i_3 j_3}^3 - E_{i_3 j_3}^{*13}\right)^2 \sum_{i_2=1}^{n^2} \frac{T_{i_1 i_2}^{*12} T_{i_2 i_3}^{*23}}{p_{i_2}^2} \sum_{j_2=1}^{n^2} \frac{T_{j_1 j_2}^{*12} T_{j_2 j_3}^{*23}}{p_{j_2}^2}}
$$

$$
= \sqrt{\sum_{i_1,j_1=1}^{n^1} \sum_{i_2,j_2=1}^{n^2} \sum_{i_3,j_3=1}^{n^3} \left(C_{i_1 j_1}^1 - C_{i_2 j_2}^2 + C_{i_2 j_2}^2 - C_{i_3 j_3}^3 - E_{i_3 j_3}^{*13}\right)^2 \frac{T_{i_1 i_2}^{*12} T_{i_2 i_3}^{*23}}{p_{i_2}^2} \frac{T_{j_1 j_2}^{*12} T_{j_2 j_3}^{*23}}{p_{j_2}^2}}
$$

$$
\leq \sqrt{\sum_{i_1,j_1=1}^{n^1} \sum_{i_2,j_2=1}^{n^2} \sum_{i_3,j_3=1}^{n^3} \left(C_{i_1 j_1}^1 - C_{i_2 j_2}^2\right)^2 \frac{T_{i_1 i_2}^{*12} T_{i_2 i_3}^{*23}}{p_{i_2}^2} \frac{T_{j_1 j_2}^{*12} T_{j_2 j_3}^{*23}}{p_{j_2}^2}}
$$

$$
+ \sqrt{\sum_{i_1,j_1=1}^{n^1} \sum_{i_2,j_2=1}^{n^2} \sum_{i_3,j_3=1}^{n^3} \left(C_{i_2 j_2}^2 - C_{i_3 j_3}^3 - E_{i_3 j_3}^{*13}\right)^2 \frac{T_{i_1 i_2}^{*12} T_{i_2 i_3}^{*23}}{p_{i_2}^2} \frac{T_{j_1 j_2}^{*12} T_{j_2 j_3}^{*23}}{p_{j_2}^2}}
$$

$$
= \sqrt{\sum_{i_1,j_1=1}^{n^1} \sum_{i_2,j_2=1}^{n^2} \left(C_{i_1 j_1}^1 - C_{i_2 j_2}^2\right)^2 T_{i_1 i_2}^{*12} T_{j_1 j_2}^{*12}} + \sqrt{\sum_{i_2,j_2=1}^{n^2} \sum_{i_3,j_3=1}^{n^3} \left(C_{i_2 j_2}^2 - C_{i_3 j_3}^3 - E_{i_3 j_3}^{*13}\right)^2 T_{i_2 i_3}^{*23} T_{j_2 j_3}^{*23}}
$$

$$
\leq \mathrm{RGWD}\left((\mathbf{C}^1, \mathbf{p}^1), (\mathbf{C}^2, \mathbf{p}^2), \epsilon\right) + \mathrm{RGWD}\left((\mathbf{C}^2, \mathbf{p}^2), (\mathbf{C}^3, \mathbf{p}^3), \epsilon\right).
$$

(**iv**) Finally, we prove the invariance to the node order permutation. Denote the solution to the objective of RGWD by $\mathbf{T}^* = [T_{ii'}^*]$ and $\mathbf{E}^* = [E_{i'j'}^*]$, which implies

$$E_{i'j'}^* = \begin{cases} \epsilon, & \text{if } \sum_{i,j=1}^{n^{\mathrm{s}}} T_{ii'}^* T_{jj'}^* (C_{ij}^{\mathrm{s}} - C_{i'j'}^{\mathrm{t}}) \leq 0, \\ -\epsilon, & \text{otherwise,} \end{cases}$$

and

$$\sum_{i,j=1}^{n^{\mathrm{s}}} \sum_{i',j'=1}^{n^{\mathrm{t}}} (C_{ij}^{\mathrm{s}} - C_{i'j'}^{\mathrm{t}} - E_{i'j'}^*)^2 T_{ii'}^* T_{jj'}^* \leq \max_{\mathbf{E} \in \mathcal{U}_\epsilon} \sum_{i,j=1}^{n^{\mathrm{s}}} \sum_{i',j'=1}^{n^{\mathrm{t}}} (C_{ij}^{\mathrm{s}} - C_{i'j'}^{\mathrm{t}} - E_{i'j'})^2 T_{ii'} T_{jj'},$$

for all $\mathbf{T} \in \Pi(\mathbf{p}^{\mathrm{s}}, \mathbf{p}^{\mathrm{t}})$. The two permutation operations can be equivalently denoted as two bijectives $\pi^{\mathrm{s}}$ and $\pi^{\mathrm{t}}$. Denote

$$\tilde{\mathbf{C}}^{\mathrm{s}} = [\tilde{C}_{ij}^{\mathrm{s}}] \text{ where } \tilde{C}_{ij}^{\mathrm{s}} = C_{\pi^{\mathrm{s}-1}(i)\pi^{\mathrm{s}-1}(j)}^{\mathrm{s}},$$

$$\tilde{\mathbf{C}}^{\mathrm{t}} = [\tilde{C}_{i'j'}^{\mathrm{t}}] \text{ where } \tilde{C}_{i'j'}^{\mathrm{t}} = C_{\pi^{\mathrm{t}-1}(i')\pi^{\mathrm{t}-1}(j')}^{\mathrm{t}},$$

$$\tilde{\mathbf{E}}^* = [\tilde{E}_{i'j'}^*] \text{ where } \tilde{E}_{i'j'}^* = E_{\pi^{\mathrm{t}-1}(i')\pi^{\mathrm{t}-1}(j')}^*,$$

$$\tilde{\mathbf{T}}^* = [\tilde{T}_{ii'}^{\mathrm{s}}] \text{ where } \tilde{T}_{ii'}^{\mathrm{s}} = T_{\pi^{\mathrm{s}-1}(i)\pi^{\mathrm{t}-1}(i')}^*.$$

We first prove $\tilde{\mathbf{E}}^*$ solves the inner maximization problem for $\tilde{\mathbf{T}}^*$. For all $i', j' \in [\![n^{\mathrm{t}}]\!]$, when $\sum_{ij} \tilde{T}_{ii'}^* \tilde{T}_{jj'}^* (\tilde{C}_{ij}^{\mathrm{s}} - \tilde{C}_{i'j'}^{\mathrm{t}}) \leq 0$, we have $\sum_{ij} T_{i\pi^{\mathrm{t}-1}(i')}^* T_{j\pi^{\mathrm{t}-1}(j')}^* (C_{ij}^{\mathrm{s}} - C_{\pi^{\mathrm{t}-1}(i')\pi^{\mathrm{t}-1}(j')}^{\mathrm{t}}) \leq 0$, which is consistent with $\tilde{E}_{i'j'}^* = \epsilon$. The case when $\sum_{ij} \tilde{T}_{ii'}^* \tilde{T}_{jj'}^* (\tilde{C}_{ij}^{\mathrm{s}} - \tilde{C}_{i'j'}^{\mathrm{t}}) > 0$ is similar. Since

$$\sum_{i,j=1}^{n^{\mathrm{s}}} \sum_{i',j'=1}^{n^{\mathrm{t}}} (C_{ij}^{\mathrm{s}} - C_{i'j'}^{\mathrm{t}} - E_{i'j'}^*)^2 T_{ii'}^* T_{jj'}^* = \sum_{i,j=1}^{n^{\mathrm{s}}} \sum_{i',j'=1}^{n^{\mathrm{t}}} (\tilde{C}_{ij}^{\mathrm{s}} - \tilde{C}_{i'j'}^{\mathrm{t}} - \tilde{E}_{i'j'}^*)^2 \tilde{T}_{ii'}^* \tilde{T}_{jj'}^*$$

and

$$\max_{\mathbf{E} \in \mathcal{U}_\epsilon} \sum_{i,j=1}^{n^{\mathrm{s}}} \sum_{i',j'=1}^{n^{\mathrm{t}}} (C_{ij}^{\mathrm{s}} - C_{i'j'}^{\mathrm{t}} - E_{i'j'})^2 T_{ii'} T_{jj'} = \max_{\mathbf{E} \in \mathcal{U}_\epsilon} \sum_{i,j=1}^{n^{\mathrm{s}}} \sum_{i',j'=1}^{n^{\mathrm{t}}} (\tilde{C}_{ij}^{\mathrm{s}} - \tilde{C}_{i'j'}^{\mathrm{t}} - E_{i'j'})^2 \tilde{T}_{ii'} \tilde{T}_{jj'},$$

where $\tilde{T}_{ii'} = T_{\pi^{\mathrm{s}-1}(i)\pi^{\mathrm{t}-1}(i')}$, $\tilde{\mathbf{T}}^*$ and $\tilde{\mathbf{E}}^*$ solve the optimization problem of RGWD $\left((\mathbf{Q}^{\mathrm{s}\top} \mathbf{C}^{\mathrm{s}} \mathbf{Q}^{\mathrm{s}}, \mathbf{Q}^{\mathrm{s}\top} \mathbf{p}^{\mathrm{s}}), (\mathbf{Q}^{\mathrm{t}\top} \mathbf{C}^{\mathrm{t}} \mathbf{Q}^{\mathrm{t}}, \mathbf{Q}^{\mathrm{t}\top} \mathbf{p}^{\mathrm{t}}), \epsilon\right)$.

∎

To prove Theorem 2, we require the following lemma.

**Lemma 3** $f(\cdot)$ *is l-smooth and L-Lipschitz, where* $l = \sqrt{2} \max\{10n^3 U_1^2 + 6n^3 U_1 \epsilon + 4n U_1 U_2 + 4n^3 \epsilon^2, 6n^2 U_1 U_2 + 2U_2^2 + 4n^2 U_2 \epsilon\}$ *and* $L = \sqrt{2} \max\{(4U_1 + 2\epsilon)U_2^2 n^3, 2(2U_1 + \epsilon)^2 U_2 n^3\}$ *with* $n = \max\{n^{\mathrm{s}}, n^{\mathrm{t}}\}$, $U_1 = \max\{\|\mathbf{C}^{\mathrm{s}}\|_\infty, \|\mathbf{C}^{\mathrm{t}}\|_\infty\}$ *and* $U_2 = \max\{\|\mathbf{p}^{\mathrm{t}}\|_2, \max_{\mathbf{T}' \in \Pi(\mathbf{p}^{\mathrm{s}}, \mathbf{p}^{\mathrm{t}})} \|\mathbf{T}'\|_F\}$.

**Proof:** (**i**) We first prove that $f(\cdot)$ is $L$-Lipschitz. For all $\mathbf{T}, \mathbf{T}' \in \Pi(\mathbf{p}^{\mathrm{s}}, \mathbf{p}^{\mathrm{t}})$ and $\mathbf{E}, \mathbf{E}' \in \mathcal{U}_\epsilon$,

$$\left| f(\mathbf{T}, \mathbf{E}) - f(\mathbf{T}', \mathbf{E}') \right|$$

$$\leq \left| \sum_{iji'j'} (C_{ij}^{\mathrm{s}} - C_{i'j'}^{\mathrm{t}} - E_{i'j'})^2 T_{ii'} T_{jj'} - \sum_{iji'j'} (C_{ij}^{\mathrm{s}} - C_{i'j'}^{\mathrm{t}} - E_{i'j'}')^2 T_{ii'} T_{jj'} \right|$$

$$+ \left| \sum_{iji'j'} (C_{ij}^{\mathrm{s}} - C_{i'j'}^{\mathrm{t}} - E_{i'j'}')^2 T_{ii'} T_{jj'} - \sum_{iji'j'} (C_{ij}^{\mathrm{s}} - C_{i'j'}^{\mathrm{t}} - E_{i'j'}')^2 T_{ii'} T_{jj'}' \right|$$

$$+ \left| \sum_{iji'j'} (C_{ij}^{\mathrm{s}} - C_{i'j'}^{\mathrm{t}} - E_{i'j'}')^2 T_{ii'} T_{jj'}' - \sum_{iji'j'} (C_{ij}^{\mathrm{s}} - C_{i'j'}^{\mathrm{t}} - E_{i'j'}')^2 T_{ii'}' T_{jj'}' \right|.$$

For the first term,

$$
\Big| \sum_{iji'j'} (C_{ij}^{\mathrm{s}} - C_{i'j'}^{\mathrm{t}} - E_{i'j'})^2 T_{ii'} T_{jj'} - \sum_{iji'j'} (C_{ij}^{\mathrm{s}} - C_{i'j'}^{\mathrm{t}} - E'_{i'j'})^2 T_{ii'} T_{jj'} \Big|
$$

$$
= \Big| \sum_{iji'j'} (C_{ij}^{\mathrm{s}} - C_{i'j'}^{\mathrm{t}} - E_{i'j'} + C_{ij}^{\mathrm{s}} - C_{i'j'}^{\mathrm{t}} - E'_{i'j'})(E'_{i'j'} - E_{i'j'}) T_{ii'} T_{jj'} \Big|
$$

$$
\leq \sum_{iji'j'} \Big| C_{ij}^{\mathrm{s}} - C_{i'j'}^{\mathrm{t}} - E_{i'j'} + C_{ij}^{\mathrm{s}} - C_{i'j'}^{\mathrm{t}} - E'_{i'j'} \Big| \Big| E'_{i'j'} - E_{i'j'} \Big| T_{ii'} T_{jj'}
$$

$$
\leq (4U_1 + 2\epsilon) U_2^2 n^2 \sum_{i'j'} \Big| E'_{i'j'} - E_{i'j'} \Big|.
$$

For the second term,

$$
\Big| \sum_{iji'j'} (C_{ij}^{\mathrm{s}} - C_{i'j'}^{\mathrm{t}} - E'_{i'j'})^2 T_{ii'} T_{jj'} - \sum_{iji'j'} (C_{ij}^{\mathrm{s}} - C_{i'j'}^{\mathrm{t}} - E'_{i'j'})^2 T_{ii'} T'_{jj'} \Big|
$$

$$
\leq \sum_{iji'j'} \Big| C_{ij}^{\mathrm{s}} - C_{i'j'}^{\mathrm{t}} - E'_{i'j'} \Big|^2 \Big| T_{ii'} \Big| \Big| T_{jj'} - T'_{jj'} \Big|
$$

$$
\leq (2U_1 + \epsilon)^2 U_2 \sum_{iji'j'} \Big| T_{jj'} - T'_{jj'} \Big|
$$

$$
\leq (2U_1 + \epsilon)^2 U_2 n^2 \Big| T_{jj'} - T'_{jj'} \Big|.
$$

For the third term,

$$
\Big| \sum_{iji'j'} (C_{ij}^{\mathrm{s}} - C_{i'j'}^{\mathrm{t}} - E'_{i'j'})^2 T_{ii'} T'_{jj'} - \sum_{iji'j'} (C_{ij}^{\mathrm{s}} - C_{i'j'}^{\mathrm{t}} - E'_{i'j'})^2 T'_{ii'} T'_{jj'} \Big|
$$

$$
\leq (2U_1 + \epsilon)^2 U_2 n^2 \Big| T_{jj'} - T'_{jj'} \Big|.
$$

Combining the three relations above, we have

$$
\big| f(\mathbf{T}, \mathbf{E}) - f(\mathbf{T}', \mathbf{E}') \big|
$$

$$
\leq \max \Big\{ (4U_1 + 2\epsilon) U_2^2 n^2, (2U_1 + \epsilon)^2 U_2 n^2 \Big\} \Big( \sum_{i'j'} |E'_{i'j'} - E_{i'j'}| + \sum_{jj'} |T_{jj'} - T'_{jj'}| \Big)
$$

$$
\leq L \sqrt{\|\mathbf{E} - \mathbf{E}'\|_F^2 + \|\mathbf{T} - \mathbf{T}'\|_F^2}.
$$

(ii) Now we prove that $f(\cdot)$ is $l$-smooth, which requires finding a constant $l$ satisfying

$$
\left\| \begin{bmatrix} \mathrm{vec}\left(\nabla_{\mathbf{T}} f(\mathbf{T}, \mathbf{E})\right) \\ \mathrm{vec}\left(\nabla_{\mathbf{E}} f(\mathbf{T}, \mathbf{E})\right) \end{bmatrix} - \begin{bmatrix} \mathrm{vec}\left(\nabla_{\mathbf{T}} f(\mathbf{T}', \mathbf{E}')\right) \\ \mathrm{vec}\left(\nabla_{\mathbf{E}} f(\mathbf{T}', \mathbf{E}')\right) \end{bmatrix} \right\|_2 \leq l \left\| \begin{bmatrix} \mathrm{vec}(\mathbf{T}) \\ \mathrm{vec}(\mathbf{E}) \end{bmatrix} - \begin{bmatrix} \mathrm{vec}(\mathbf{T}') \\ \mathrm{vec}(\mathbf{E}') \end{bmatrix} \right\|_2,
$$

where $\mathrm{vec}(\mathbf{X})$ means the vectorization of matrix $\mathbf{X}$ and $\begin{bmatrix} \mathbf{a} \\ \mathbf{b} \end{bmatrix}$ denotes the concatenation of vectors $\mathbf{a}$ and $\mathbf{b}$. Since the left hand side satisfies

$$
\left\| \begin{bmatrix} \mathrm{vec}\left(\nabla_{\mathbf{T}} f(\mathbf{T}, \mathbf{E})\right) \\ \mathrm{vec}\left(\nabla_{\mathbf{E}} f(\mathbf{T}, \mathbf{E})\right) \end{bmatrix} - \begin{bmatrix} \mathrm{vec}\left(\nabla_{\mathbf{T}} f(\mathbf{T}', \mathbf{E}')\right) \\ \mathrm{vec}\left(\nabla_{\mathbf{E}} f(\mathbf{T}', \mathbf{E}')\right) \end{bmatrix} \right\|_2
$$

$$
= \sqrt{\|\nabla_{\mathbf{T}} f(\mathbf{T}, \mathbf{E}) - \nabla_{\mathbf{T}} f(\mathbf{T}', \mathbf{E}')\|_F^2 + \|\nabla_{\mathbf{E}} f(\mathbf{T}, \mathbf{E}) - \nabla_{\mathbf{E}} f(\mathbf{T}', \mathbf{E}')\|_F^2}
$$

$$
\leq \|\nabla_{\mathbf{T}} f(\mathbf{T}, \mathbf{E}) - \nabla_{\mathbf{T}} f(\mathbf{T}', \mathbf{E}')\|_F + \|\nabla_{\mathbf{E}} f(\mathbf{T}, \mathbf{E}) - \nabla_{\mathbf{E}} f(\mathbf{T}', \mathbf{E}')\|_F,
$$

and the right hand side satisfies

$$
l \left\| \begin{bmatrix} \mathrm{vec}(\mathbf{T}) \\ \mathrm{vec}(\mathbf{E}) \end{bmatrix} - \begin{bmatrix} \mathrm{vec}(\mathbf{T}') \\ \mathrm{vec}(\mathbf{E}') \end{bmatrix} \right\|_2 = l \sqrt{\|\mathbf{E} - \mathbf{E}'\|_F^2 + \|\mathbf{T} - \mathbf{T}'\|_F^2}
$$

$$
\geq \frac{l}{\sqrt{2}} \|\mathbf{E} - \mathbf{E}'\|_F + \frac{l}{\sqrt{2}} \|\mathbf{T} - \mathbf{T}'\|_F,
$$

it suffices to find a constant $l$ satisfying

$$\|\nabla_{\mathbf{T}}f(\mathbf{T},\mathbf{E})-\nabla_{\mathbf{T}}f(\mathbf{T}',\mathbf{E}')\|_F+\|\nabla_{\mathbf{E}}f(\mathbf{T},\mathbf{E})-\nabla_{\mathbf{E}}f(\mathbf{T}',\mathbf{E}')\|_F \le \frac{l}{\sqrt{2}}\|\mathbf{E}-\mathbf{E}'\|_F+\frac{l}{\sqrt{2}}\|\mathbf{T}-\mathbf{T}'\|_F.$$

We bound $\|\nabla_{\mathbf{T}}f(\mathbf{T},\mathbf{E})-\nabla_{\mathbf{T}}f(\mathbf{T}',\mathbf{E}')\|_F$ as follows,

$$\|\nabla_{\mathbf{T}}f(\mathbf{T},\mathbf{E})-\nabla_{\mathbf{T}}f(\mathbf{T}',\mathbf{E}')\|_F$$
$$\le 2n\|\mathbf{C}^{\mathrm{s}}\odot\mathbf{C}^{\mathrm{s}}\|_F\|\mathbf{T}-\mathbf{T}'\|_F+4\|\mathbf{C}^{\mathrm{s}}\|_F\|\mathbf{T}(\mathbf{C}^{\mathrm{t}}+\mathbf{E})-\mathbf{T}'(\mathbf{C}^{\mathrm{t}}+\mathbf{E}')\|_F$$
$$+2n\|\mathbf{T}(\mathbf{C}^{\mathrm{t}}+\mathbf{E})\odot(\mathbf{C}^{\mathrm{t}}+\mathbf{E})-\mathbf{T}'(\mathbf{C}^{\mathrm{t}}+\mathbf{E}')\odot(\mathbf{C}^{\mathrm{t}}+\mathbf{E}')\|_F.$$

For the first term,

$$2n\|\mathbf{C}^{\mathrm{s}}\odot\mathbf{C}^{\mathrm{s}}\|_F\|\mathbf{T}-\mathbf{T}'\|_F \le 2n\|\mathbf{C}^{\mathrm{s}}\|_F^2 \le 2n^3U_1^2\|\mathbf{T}-\mathbf{T}'\|_F.$$

For the second term,

$$4\|\mathbf{C}^{\mathrm{s}}\|_F\|\mathbf{T}(\mathbf{C}^{\mathrm{t}}+\mathbf{E})-\mathbf{T}'(\mathbf{C}^{\mathrm{t}}+\mathbf{E}')\|_F$$
$$\le 4\|\mathbf{C}^{\mathrm{s}}\|_F\|\mathbf{T}-\mathbf{T}'\|_F\Big(\|\mathbf{C}^{\mathrm{t}}\|_F+\|\mathbf{E}\|_F\Big)+4\|\mathbf{C}^{\mathrm{s}}\|_F\|\mathbf{T}'\|_F\|\mathbf{E}-\mathbf{E}'\|_F$$
$$\le(4n^2U_1^2+4n^2U_1\epsilon)\|\mathbf{T}-\mathbf{T}'\|_F+4nU_1U_2\|\mathbf{E}-\mathbf{E}'\|_F.$$

For the third term,

$$2n\|\mathbf{T}(\mathbf{C}^{\mathrm{t}}+\mathbf{E})\odot(\mathbf{C}^{\mathrm{t}}+\mathbf{E})-\mathbf{T}'(\mathbf{C}^{\mathrm{t}}+\mathbf{E}')\odot(\mathbf{C}^{\mathrm{t}}+\mathbf{E}')\|_F$$
$$\le 2n\|\mathbf{T}(\mathbf{C}^{\mathrm{t}}+\mathbf{E})\odot(\mathbf{C}^{\mathrm{t}}+\mathbf{E})-\mathbf{T}'(\mathbf{C}^{\mathrm{t}}+\mathbf{E})\odot(\mathbf{C}^{\mathrm{t}}+\mathbf{E})\|_F$$
$$+2n\|\mathbf{T}'(\mathbf{C}^{\mathrm{t}}+\mathbf{E})\odot(\mathbf{C}^{\mathrm{t}}+\mathbf{E})-\mathbf{T}'(\mathbf{C}^{\mathrm{t}}+\mathbf{E}')\odot(\mathbf{C}^{\mathrm{t}}+\mathbf{E})\|_F$$
$$+2n\|\mathbf{T}'(\mathbf{C}^{\mathrm{t}}+\mathbf{E}')\odot(\mathbf{C}^{\mathrm{t}}+\mathbf{E})-\mathbf{T}'(\mathbf{C}^{\mathrm{t}}+\mathbf{E}')\odot(\mathbf{C}^{\mathrm{t}}+\mathbf{E}')\|_F$$
$$\le 4n\big(\|\mathbf{C}^{\mathrm{t}}\|_F^2+\|\mathbf{E}\|_F^2\big)\|\mathbf{T}-\mathbf{T}'\|_F+2n\|\mathbf{T}'\|_F\big(\|\mathbf{C}^{\mathrm{t}}\|_F+\|\mathbf{E}\|_F\big)\|\mathbf{E}-\mathbf{E}'\|_F$$
$$+2n\|\mathbf{T}'\|_F\big(\|\mathbf{C}^{\mathrm{t}}\|_F+\|\mathbf{E}'\|_F\big)\|\mathbf{E}-\mathbf{E}'\|_F$$
$$\le\big(4n^3U_1^2+4n^3\epsilon^2\big)\|\mathbf{T}-\mathbf{T}'\|_F+(4n^2U_1U_2+4n^2U_2\epsilon)\|\mathbf{E}-\mathbf{E}'\|_F$$

Since $\nabla_{\mathbf{E}}f(\mathbf{T},\mathbf{E})=2(\mathbf{E}+\mathbf{C}^\top)\mathbf{p}^{\mathrm{t}}\mathbf{p}^{\mathrm{t}\,\top}-2\mathbf{T}^\top\mathbf{C}^{\mathrm{s}}\mathbf{T},\|\nabla_{\mathbf{E}}f(\mathbf{T},\mathbf{E})-\nabla_{\mathbf{E}}f(\mathbf{T}',\mathbf{E}')\|_F$ can be bounded as follows,

$$\|\nabla_{\mathbf{E}}f(\mathbf{T},\mathbf{E})-\nabla_{\mathbf{E}}f(\mathbf{T}',\mathbf{E}')\|_F$$
$$\le\|2(\mathbf{E}-\mathbf{E}')\|_F\|\mathbf{p}^{\mathrm{t}}\|_F^2+4\|\mathbf{T}\|_F\|\mathbf{C}^{\mathrm{s}}\|_F\|\mathbf{T}-\mathbf{T}'\|_F$$
$$\le 2U_2^2\|(\mathbf{E}-\mathbf{E}')\|_F+4nU_1U_2\|\mathbf{T}-\mathbf{T}'\|_F$$
$$.$$

Combining the above four relations, we have

$$\|\nabla_{\mathbf{T}}f(\mathbf{T},\mathbf{E})-\nabla_{\mathbf{T}}f(\mathbf{T}',\mathbf{E}')\|_F+\|\nabla_{\mathbf{E}}f(\mathbf{T},\mathbf{E})-\nabla_{\mathbf{E}}f(\mathbf{T}',\mathbf{E}')\|_F$$
$$\le\max\{10n^3U_1^2+6n^3U_1\epsilon+4nU_1U_2+4n^3\epsilon^2,6n^2U_1U_2+2U_2^2+4n^2U_2\epsilon\}\Big(\|\mathbf{E}-\mathbf{E}'\|_F+\|\mathbf{T}-\mathbf{T}'\|_F\Big),$$

which yields the desired result.

∎

**Theorem 2** *Define $\phi(\cdot)=\max_{\mathbf{E}\in\mathcal{U}_\epsilon}f(\cdot,\mathbf{E})$. The output $\hat{\mathbf{T}}$ of Algorithm 1 with step-size $\eta=\frac{\gamma}{\sqrt{N+1}}$ satisfies*

$$\mathbb{E}\big[\|\nabla\phi_{1/2l}(\hat{\mathbf{T}})\|^2\big]\le 2\frac{\phi_{1/2l}(\mathbf{T}_0)-\min_{\mathbf{T}\in\Pi(\mathbf{p}^{\mathrm{s}},\mathbf{p}^{\mathrm{t}})}\phi(\mathbf{T})+lL^2\gamma^2}{\gamma\sqrt{N+1}},$$

*where $l=\sqrt{2}\max\{10n^3U_1^2+6n^3U_1\epsilon+4nU_1U_2+4n^3\epsilon^2,6n^2U_1U_2+2U_2^2+4n^2U_2\epsilon\}$ and $L=\sqrt{2}\max\{(4U_1+2\epsilon)U_2^2n^3,2(2U_1+\epsilon)^2U_2n^3\}$ with $n=\max\{n^{\mathrm{s}},n^{\mathrm{t}}\}$, $U_1=\max\{\|\mathbf{C}^{\mathrm{s}}\|_\infty,\|\mathbf{C}^{\mathrm{t}}\|_\infty\}$ and $U_2=\max\{\|\mathbf{p}^{\mathrm{t}}\|_2,\max_{\mathbf{T}'\in\Pi(\mathbf{p}^{\mathrm{s}},\mathbf{p}^{\mathrm{t}})}\|\mathbf{T}'\|_F\}$.*

**Proof:** By the smoothness of $f(\cdot)$, for any $\tilde{\mathbf{T}} \in \Pi(\mathbf{p}^s, \mathbf{p}^t)$ and $\mathbf{T}_\tau$ from Algorithm 1, we have

$$\phi(\tilde{\mathbf{T}}) \geq f(\tilde{\mathbf{T}}, \mathbf{E}_\tau) \geq f(\mathbf{T}_\tau, \mathbf{E}_\tau) + \langle \nabla_{\mathbf{T}} f(\mathbf{T}_\tau, \mathbf{E}_\tau), \tilde{\mathbf{T}} - \mathbf{T}_\tau \rangle - \frac{l}{2} \|\tilde{\mathbf{T}} - \mathbf{T}_\tau\|_F^2$$

$$= \phi(\mathbf{T}_\tau) + \langle \nabla_{\mathbf{T}} f(\mathbf{T}_\tau, \mathbf{E}_\tau), \tilde{\mathbf{T}} - \mathbf{T}_\tau \rangle - \frac{l}{2} \|\tilde{\mathbf{T}} - \mathbf{T}_\tau\|_F^2. \tag{6}$$

Let $\hat{\mathbf{T}}_\tau = \operatorname{argmin}_{\mathbf{T} \in \Pi(\mathbf{p}^s, \mathbf{p}^t)} \phi(\mathbf{T}) + l\|\mathbf{T} - \mathbf{T}_\tau\|_F^2$. We have

$$\phi_{1/2l}(\mathbf{T}_{\tau+1}) \leq \phi(\hat{\mathbf{T}}_\tau) + l\|\mathbf{T}_{\tau+1} - \hat{\mathbf{T}}_\tau\|_F^2$$

$$\leq \phi(\hat{\mathbf{T}}_\tau) + l\|\mathbf{T}_\tau - \eta\nabla_{\mathbf{T}} f(\mathbf{T}_\tau, \mathbf{E}_\tau) - \hat{\mathbf{T}}_\tau\|_F^2$$

$$\leq \phi(\hat{\mathbf{T}}_\tau) + l\|\mathbf{T}_\tau - \hat{\mathbf{T}}_\tau\|_F^2 + 2l\eta\langle\nabla_{\mathbf{T}} f(\mathbf{T}_\tau, \mathbf{E}_\tau), \hat{\mathbf{T}}_\tau - \mathbf{T}_\tau\rangle + \eta^2 l\|\nabla_{\mathbf{T}} f(\mathbf{T}_\tau, \mathbf{E}_\tau)\|_F^2$$

$$\leq \phi_{1/2l}(\mathbf{T}_\tau) + 2l\eta\langle\nabla_{\mathbf{T}} f(\mathbf{T}_\tau, \mathbf{E}_\tau), \hat{\mathbf{T}}_\tau - \mathbf{T}_\tau\rangle + \eta^2 l\|\nabla_{\mathbf{T}} f(\mathbf{T}_\tau, \mathbf{E}_\tau)\|_F^2$$

$$\leq \phi_{1/2l}(\mathbf{T}_\tau) + 2\eta l\Big(\phi(\hat{\mathbf{T}}_\tau) - \phi(\mathbf{T}_\tau) + \frac{l}{2}\|\hat{\mathbf{T}}_\tau - \mathbf{T}_\tau\|_F^2\Big) + \eta^2 lL^2,$$

where the second line uses Lemma 3.1 of Bubeck et al. (2015) and the last line follows from (6). Taking a telescopic sum over $\tau$, we obtain

$$\phi_{1/2l}(\mathbf{T}_N) \leq \phi_{1/2l}(\mathbf{T}_0) + 2\eta l\sum_{\tau=0}^{N}\Big(\phi(\hat{\mathbf{T}}_\tau) - \phi(\mathbf{T}_\tau) + \frac{l}{2}\|\hat{\mathbf{T}}_\tau - \mathbf{T}_\tau\|_F^2\Big) + \eta^2 lL^2.$$

Rearranging this, we obtain

$$\frac{1}{N+1}\sum_{\tau=0}^{N}\Big(-\phi(\hat{\mathbf{T}}_\tau) + \phi(\mathbf{T}_\tau - \frac{l}{2}\|\hat{\mathbf{T}}_\tau - \mathbf{T}_\tau\|_F^2\Big) \leq \frac{\phi_{1/2l}(\mathbf{T}_0) - \min_{\mathbf{T}\in\Pi(\mathbf{p}^s,\mathbf{p}^t)}\phi(\mathbf{T})}{2\eta lN} + \frac{\eta L^2}{2}. \tag{7}$$

Since $\phi(\mathbf{T}) + l\|\mathbf{T} - \mathbf{T}_\tau\|_F^2$ is $l$-strongly convex, we have

$$-\phi(\hat{\mathbf{T}}_\tau) + \phi(\mathbf{T}_\tau) - \frac{l}{2}\|\hat{\mathbf{T}}_\tau - \mathbf{T}_\tau\|_F^2$$

$$\geq \frac{l}{2}\|\mathbf{T}_\tau - \hat{\mathbf{T}}_\tau\|_F^2 + \phi(\mathbf{T}_\tau) + l\|\mathbf{T}_\tau - \mathbf{T}_\tau\|_F^2 - \min_{\mathbf{T}}\Big(\phi(\mathbf{T}) + l\|\mathbf{T} - \mathbf{T}_\tau\|_F^2\Big)$$

$$\geq l\|\mathbf{T}_\tau - \hat{\mathbf{T}}_\tau\|_F^2 = \frac{1}{4l}\|\nabla\phi_{1/2l}(\mathbf{T}_\tau)\|_F^2.$$

Plugging this in (7) and combining Lemma 3 proves the result.

■

# B ALGORITHMIC DETAILS

The Projected Gradient Descent (PGD) consists of the following three steps in each iteration $\tau$.

**Find $\mathbf{E}_\tau$ that maximizes $f(\mathbf{T}_\tau, \mathbf{E})$.** By Theorem 1, we need to calculate an auxiliary matrix

$$\mathbf{G} = \mathbf{T}_\tau^\top \mathbf{C}^s \mathbf{T}_\tau - \mathbf{C}^t \odot \mathbf{T}_\tau^\top \mathbf{T}_\tau,$$

where $\odot$ denotes the element-wise multiplication. And we have

$$E_{i'j'}(\mathbf{T}_\tau) = \begin{cases} \epsilon, & \text{if } G_{i'j'} \leq 0, \\ -\epsilon, & \text{otherwise.} \end{cases}$$

Such a step involves computational cost $\mathcal{O}(n^3)$ where $n = \max\{n^s, n^t\}$.

**Gradient descent.** Calculate $\mathbf{H}_\tau = \mathbf{T}_\tau - \eta\nabla_{\mathbf{T}} f(\mathbf{T}_\tau, \mathbf{E}_\tau)$ where

$$\nabla_{\mathbf{T}} f(\mathbf{T}_\tau, \mathbf{E}_\tau) = 2(\mathbf{C}^s \odot \mathbf{C}^s)\mathbf{T}_\tau \mathbf{1}^{n^t}\mathbf{1}^{n^t\top} + 2\mathbf{1}^{n^s}\mathbf{1}^{n^s\top}\mathbf{T}_\tau\Big((\mathbf{C}^t + \mathbf{E}_\tau) \odot (\mathbf{C}^t + \mathbf{E}_\tau)\Big) - 4\mathbf{C}^s\mathbf{T}_\tau(\mathbf{C}^t + \mathbf{E}_\tau),$$

which also involves computational cost $\mathcal{O}(n^3)$.

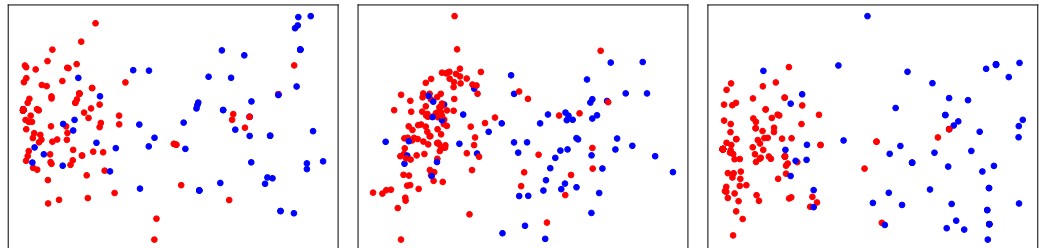

Figure 2: PCA-based visualization of embeddings produced by GDL (left), GWF (middle), and RGDL (right) respectively for the graphs in MUTAG dataset. The color of each point indicate the type of the corresponding graph. RGDL achieves the best clustering results.

**Projection into the feasible domain.** This requires solving the following problem

$$\min_{\mathbf{T} \geq 0} \frac{1}{2}\|\mathbf{T} - \mathbf{H}_\tau\|_F^2, \text{ s.t. } \mathbf{T}\mathbf{1}^n = \mathbf{p}, \mathbf{T}^\top\mathbf{1}^m = \mathbf{q}.$$

This optimization problem has a strongly convex objective and linear constraints, and hence can be solved efficiently via Augmented Lagrangian Method with computational complexity $\frac{n^2|\log\rho|}{\rho^{1/2}}$ (Xu, 2021) where $\rho$ measures the optimality, that is, the violation of the two linear constraints. When $\rho = \mathcal{O}\left(\frac{1}{n^2}\right)$, this step also has cubic costs if we ignore the log term.

Therefore, the overall complexity of PGD obtaining a $\delta$-stationary solution is $\mathcal{O}\left(\frac{n^3}{\delta^2} + \frac{n^2|\log\rho|}{\rho^{1/2}\delta^2}\right)$.

## C  ADDITIONAL EXPERIMENTS

### C.1  ADDITIONAL EXPERIMENTAL RESULTS OF GRAPH CLUSTERING

**Visualization of graph embeddings.** Since GDL, GWF, and RGDL can output graph embeddings, we further illustrate the embeddings generated by them respectively based on PCA. As is shown in Figure 2, the embeddings of the two types of the graphs are less likely to be mixed together, which explains why RGDL achieves higher ARI values.

**Sensitivity analysis of $\lambda$.** As is discussed in Li et al. (2020); Vincent-Cuaz et al. (2021), the negative quadratic term can promote the sparsity of graph embeddings. We further conduct sensitivity analysis of $\lambda$ by varying the value in $\{0, 10^{-5}, 10^{-4}, 10^{-3}, 10^{-2}, 10^{-1}\}$. As is shown in Table 3, $\lambda \in [10^{-4}, 10^{-2}]$ often yields good performance. The experiments in the main paper are run with $\lambda = 10^{-3}$.

Table 3: ARI scores of RGDL with varied $\lambda$'s.

| Datasets | 0 | $10^{-5}$ | $10^{-4}$ | $10^{-3}$ | $10^{-2}$ | $10^{-1}$ |
|---|---|---|---|---|---|---|
| **MUTAG** | 0.4389 | 0.4561 | 0.4636 | 0.458 | 0.4661 | 0.4412 |
| **BZR** | 0.2515 | 0.2515 | 0.2707 | 0.2707 | 0.2707 | 0 |
| **Peking_1** | 0.1195 | 0.1195 | 0.1195 | 0.1195 | 0.1195 | 0.1079 |

### C.2  GRAPH CLASSIFICATION

The learned embeddings of graphs can also be used in the graph classification task. RGDL is thus compared against GDL (Vincent-Cuaz et al., 2021), GWF (Xu, 2020), and other state-of-the-art graph classification methods including WGDL (Zhang et al., 2021) and GNTK (Du et al., 2019) on the benchmark datasets MUTAG (Debnath et al., 1991), IMDB-B, and IMDB-M (Yanardag and Vishwanathan, 2015). RGDL, GDL, and GWF use 3-NN as the classifier due to its simplicity. We perform a 10-fold nested cross validation (using 9 folds for training, 1 for testing, and reporting the average accuracy of this experiment repeated 10 times) by keeping same folds across methods.

Table 4: Graph classification results.

| Datasets | RGDL | GDL | GWF | WGDL | GNTK |
|----------|------|-----|-----|------|------|
| **IMDB-B** | 77.7(1.5) | 59.0(2.6) | 53.0(6.2) | 79.7(3.6) | 76.9(3.6) |
| **IMDB-M** | 52.9(2.5) | 44.2(2.3) | 42.0(3.7) | 53.5(5.0) | 52.8(4.6) |
| **MUTAG** | 98.2(3.0) | 89.5(5.3) | 86.0(3.0) | 94.7(2.6) | 90.0(8.5) |

The results are reported in Table 4 and RGDL outperforms or matches state-of-the-art methods. RGDL outperforms GDL and GWF significantly, which indicates the necessity of taking into account the structural noise of observed graphs. Although WGDL and GNTK have similar performance, they are more computation and memory demanding due to the usage of graph neural networks.

