# OpenReview forum: "Robust Graph Dictionary Learning"
_ICLR.cc/2023/Conference — ICLR 2023 poster_

### Official Review · Reviewer_2db5 · 2022-10-23

**Confidence:** 2
**Clarity, Quality, Novelty And Reproducibility:** The quality, clarity and originality …
**Correctness:** 4
**Technical Novelty And Significance:** 3
**Empirical Novelty And Significance:** 2
**Recommendation:** 8

**Strength And Weaknesses:**

Strengths:
1. A new robust variant of GWD is proposed with discussions on its properties such as asymmetry.
2. A projected gradient descent algorithm for RGWD is provided with convergence guarantees.
3. Performance of the proposed method on synthetic and real-world datasets is outstanding.

Weaknesses:
1. In Sec 3.2, the Moreau envelope of the maximum function turns out to be a minimax problem, and the corresponding proximal operator is not straightforward. Some remarks about the role of this in Theorem 1 could be made.
2. It's not clear why a negative quadratic term is necessary in eqn.(4), which also makes the objective nonconvex and may cause convergence issues. In addition, guidance on choosing the parameter $\lambda$ for some datasets could be given.


**Summary Of The Paper:**

The paper proposes an improved graph learning algorithm based on a robust Gromov-Wasserstein discrepancy (RGWD). Following the previous GDL by Vincent-Cuaz et al. 2021, the new method replaces GWD by a new RGWD which uses a minimax formulation. Moreover, robust graph dictionary learning algorithm is also proposed based on stochastic gradient descent. Overall, the paper has a certain novelty on designing a discrepancy form with theories, and the numerical experiements on synthetic and real-world data sets have shown the performance of the proposed algorithm.

**Summary Of The Review:**

The paper is interesting, reads well and has clear motivation and contributions.

---

> ### Author Response · Authors · 2022-11-17
> **Responses to Reviewer 2db5**
>
> Thank you for taking the time to review our paper and for the invaluable comments.
>
> **Moreau Envelope in Sec. 3.2**
> Actually, since the inner maximization problem admits a closed-form solution given each transport plan, we only apply the Moreau Envelope to the outer minimization problem to derive the convergence rate of Algorithm 1.
>
> **Negative quadratic term**
> The negative quadratic term can promote sparsity for the graph embeddings, as is discussed in Vincent-Cuaz et al. [2021].
> The sensitivity analysis of $\lambda$ in Sec. C demonstrates that $\lambda\in[10^{-4},10^{-2}]$ usually yields good performance of RGDL.
> In the main paper, the empirical analysis is conducted with $\lambda=10^{-3}$.

---

> ### Author Response · Authors · 2022-12-05
> **Looking forward to your feedback**
>
> Dear Reviewer,
>
> Thank you again for your great efforts in reviewing our paper!
>
> As the deadline for the discussion is fast approaching, we would like to know whether our response addresses your concerns adequately and we would be happy to address any remaining points. Thank you so much!
>
> Sincerely,
>
> Authors

---

### Official Review · Reviewer_Lcq7 · 2022-10-24

**Confidence:** 5
**Correctness:** 3
**Technical Novelty And Significance:** 2
**Empirical Novelty And Significance:** 3
**Recommendation:** 6

**Clarity, Quality, Novelty And Reproducibility:**

The clarity and quality can be improved. The novelty is not very significant when we consider the previous work GWD and GDL. The numerical results are not sufficient.

**Strength And Weaknesses:**

Strengths:
1. Graph-level dictionary learning is a challenging problem and hasn't been well studied. The paper proposed an effective method called RGWD though heavily relying on previous work.

2. The authors provide some theoretical analysis including some properties of the proposed metric and the convergence of the optimization algorithm.

3. The numerical results especially those of the synthetic data showed the effectiveness of the proposed method in comparison to GDL.

Weaknesses:
1. In terms of the model, the contribution is not significant because compared with previous work of GWD, the proposed method only added an error term E. Thus RGWD can be regarded as a natural relaxation of GWD. The connection and difference should be highlighted. In the numerical results, what caused the improvement of RGDL over GDL? Is it the noise term E or optimization algorithm?

2. It seems that the time complexity is $O(n^4)$ and the authors did not discuss the time cost in the experiments.

3. The role and effectiveness of E are not clearly explained. Could one use other constraint on E instead of the infinity norm?

4. In Theorem 1, it is not clear why RGWD cannot reach zero.

5. Is RGWD symmetric?

6. In the experiments, are C_i similarity matrices? If yes, setting $\epsilon$ to large values such as 10 and 30 in Table 1 looks a little weird.

7. In Table 4, it seems that all methods failed in clustering IMDB-B and IMDB-M because the ARIs are less than 0.1, which makes the experimental results useless. The authors should consider more real datasets on which the proposed method can work well. One MUTAG dataset is not enough.

8. It is not clear why the authors did not show the influence of $\epsilon$ on the clustering performance of the real datasets.

**Summary Of The Paper:**

The paper studies the problem of dictionary learning for graphs. It proposed an improved graph dictionary learning algorithm based on a robust Gromov-Wasserstein discrepancy (RGWD). It also provided some theoretical results and numerical results.

**Summary Of The Review:**

Overall, the idea of robust dictionary learning is interesting though robust dictionary learning has been widely studied in many previous work of vector data. The novelty is fine and the experiments should be improved.

---

> ### Author Response · Authors · 2022-11-17
> **Responses to Reviewer Lcq7**
>
> Thank you for taking the time to review our paper and for the invaluable comments.
>
> **Significance of RGWD.**
> GWD lacks robustness, as is studied in the works that we have cited in Sec. 6.
> Instead of a natural relaxation, we seek to minimize the worst-case objective under the constraint of the perturbation, which is a prevalent approach in robust optimization [1,2].
> However, this may yield intractable nonconvex-nonconcave minimax problems.
> This paper makes the first effort towards applying robust optimization to GWD and allows comparing noisy graphs.
>
> [1] Ben-Tal, Aharon, Laurent El Ghaoui, and Arkadi Nemirovski. Robust optimization. Vol. 28. Princeton university press, 2009.
>
> [2] Rahimian, Hamed, and Sanjay Mehrotra. "Distributionally robust optimization: A review." arXiv preprint arXiv:1908.05659 (2019).
>
> **Improvement of RGDL over GDL.**
> The adoption of RGWD contributes most to the improvement of RGDL instead of the optimization algorithm.
> This can be seen from the case when $\epsilon$ is relatively small and RGDL has similar performance to GDL.
>
> **Time complexity.**
> The time complexity with respect to the number of nodes is cubic.
> The algorithmic details and the discussion of computational complexity are provided in Sec. B of the revised Appendix.
> We have also discussed the computation time in the numerical experiments by plotting ARI vs. time curves.
>
> **The role of $\mathbf{E}$.**
> $\mathbf{E}$ models the inaccuracy of the pairwise relation matrix which is restricted within a bounded set.
> Using other constraints instead of the infinity norm on $\mathbf{E}$ may also benefit the robustness, which however does not lead to the closed-form solution of the inner maximization problem.
> The resulted optimization problem is hence more difficult to solve.
> We will discuss the usage of other constraints in the long version of our paper.
>
> **Strict positiveness of RGWD.**
> RGWD is lower bounded by $\epsilon$ and the lower bound is achieved if and only if the pairwise relation matrices are identical (up to a permutation).
> Even if the compared graphs are isomorphic, since we are considering the worst-case scenario the perturbation, RGWD does not have a value of zero.
>
> **Asymmetry of RGWD.** RGWD is generally asymmetric, which is illustrated in Example 1.
>
> **C_i.**
> C_i is the heat kernel matrix of each graph as is mentioned in the first paragraph of Sec. 5.
>
> **More clustering results.**
> We have further included empirical results on the BZR and PEKING datasets in Sec. 5.2 of our revision.
> RGDL achieves ARI values greater than 0.1 on these datasets and outperforms baselines by a large margin.
>
> **Influence of $\epsilon$ on real datasets.**
> The experimental results of RGDL with varied $\epsilon$s are reported in Figure 1 of Sec. 5.2, which demonstrates that RGDL with $\epsilon\in\big[10^{-2}\max_i||\mathbf{C_i}||_\infty,10^{-1}\max_i||\mathbf{C_i}||_\infty\big]$ often achieves better results than other methods.

---

> ### Author Response · Authors · 2022-12-05
> **Looking forward to your feedback**
>
> Dear Reviewer,
>
> Thank you again for your great efforts in reviewing our paper!
>
> As the deadline for the discussion is fast approaching, we would like to know whether our response addresses your concerns adequately and we would be happy to address any remaining points. Thank you so much!
>
> Sincerely,
>
> Authors

---

### Official Review · Reviewer_ia1M · 2022-10-25

**Confidence:** 3
**Correctness:** 3
**Technical Novelty And Significance:** 2
**Empirical Novelty And Significance:** 2
**Recommendation:** 5

**Clarity, Quality, Novelty And Reproducibility:**

This paper is well-written and easy to read. It improves a previous work by adding the perturbation to the Gromov-Wasserstein metric to make it robust to noises, which is somehow marginally significant or novel. The code is provided in the supplementary material.

**Strength And Weaknesses:**

Strengths
Theoretical analysis about the properties of the proposed RGWD is provided.

Weakness
1. Lacking the motivation of using the projected gradient to update the transport plan.
2. Insufficient experimental evaluation.

Detailed comments:
1. What’s the advantage of using the projected gradient to update the transport plan.? Can it be solved by employing the Sinkhorn algorithm? More theoretical or experimental evidence needs to be provided to compare their efficiency/convergence and results.
2. Three datasets named MUTAG, IMDB-B, and IMDB-M are used to evaluate the proposed method with the metric of ARI score. However, as these datasets are usually used for graph classification, it’s suggested to provide the classification performance on these datasets as well.
3. More graph learning methods could also be compared, e.g. graph kernel-based methods, other than only graph dictionary learning. Specifically, a deep graph dictionary learning method is proposed in [a] recently. It is suggested to compare this work by discussing the advantages and disadvantages, as well as comparing the performance.
[a] T. Zhang, et al. Deep Wasserstein Graph Discriminant Learning for Graph Classification. AAAI 2021.


**Summary Of The Paper:**

This paper proposes a robust graph dictionary learning algorithm based on a metric named robust Gromov-Wasserstein discrepancy (RGWD).  With the proposed method,  a set of atoms can be learned from noisy graph data. Experiments are conducted on synthetic datasets as well as public real-world datasets to evaluate the proposed method.

**Summary Of The Review:**

The motivation of the way for optimizing the transport plan should be made clear, and more experimental results should be provided with a comprehensive comparison.

---

> ### Author Response · Authors · 2022-11-17
> **Responses to Reviewer ia1M**
>
> Thank you for taking the time to review our paper and for the invaluable comments.
>
> **Motivation of projected gradient descent (PGD).**
> PGD is a widely-adopted algorithm for constrained minimization problems due to its effectiveness and efficiency.
> Since the inner maximization problem of RGWD has a closed-form solution, RGWD can be calculated efficiently using PGD and the theoretical analysis can be extended from the constrained minimization case.
>
> **Calculating the transport plan by employing the Sinkhorn algorithm.**
> RGWD involves a quadratic programming problem and cannot be solved by the Sinkhorn algorithm.
>
> **Graph classification results.**
> RGDL outperforms or matches state-of-the-art graph classification methods including the WGDL [a] and graph kernel-based method GNTK [b], as is detailed in the revision.
> Due to the limited space of the main paper, such results are provided in Sec. C of the appendix.
>
> [a] Zhang, Tong, et al. "Deep Wasserstein Graph Discriminant Learning for Graph Classification." AAAI. 2021.
>
> [b] Du, Simon S., et al. "Graph neural tangent kernel: Fusing graph neural networks with graph kernels." Advances in neural information processing systems 32 (2019).
>
> **Comparison to WGDL**
> WGDL requires that each node is associated to an attribute vector and uses supervision to learn graph representations.
> Thus, it cannot be applied to tasks like unsupervised graph clustering.
> Our method can learn representations of unattributed graphs and conduct graph representation learning in an unsupervised manner.

---

> ### Author Response · Authors · 2022-12-05
> **Looking forward to your feedback**
>
> Dear Reviewer,
>
> Thank you again for your great efforts in reviewing our paper!
>
> As the deadline for the discussion is fast approaching, we would like to know whether our response addresses your concerns adequately and we would be happy to address any remaining points. Thank you so much!
>
> Sincerely,
>
> Authors

---

### Official Review · Reviewer_q8cY · 2022-10-26

**Confidence:** 4
**Correctness:** 4
**Technical Novelty And Significance:** 3
**Empirical Novelty And Significance:** 3
**Recommendation:** 8

**Clarity, Quality, Novelty And Reproducibility:**

The paper is clearly written and presented.

The idea appears to be novel and original, and tackles a known limitation of a very new/state of the art method.



**Strength And Weaknesses:**

- Strengths:

- The approach is well motivated and natural. The drawbacks (lack for robustness under noise) of the current GWD approach is stated, and then a solution that address the robustness drawback is proposed.

- The paper is well-written. The background material on optimal transport and graph dictionary learning is presented clearly.

- The proposed method is a very natural extension of GWD.

- Some nice (although pretty basic) properties of the RGWD, such as triangle inequality is posed.

- In experiments and simulations, RGWD appears to perform quite substantially better than competitors.

- Weaknesses:

- The comparison in the simulations is limited. In the simulations, the authors used RGWD with a wide variety of epsilon parameters and compare all of them against only GDL. While the performance of RGWD is good/better than GWD across a variety of epsilon choices, comparison against only one alternative (on a simulated graph clustering task, of which many alternatives should be available) does not convince the reader strongly enough that the RGWD is truly superior to a broad set of competitors. I highly suggest that the authors compare the performance of RGWD against a few other graph clustering methods to make their simulations more convincing.

- With any distance/divergence or other tool for graph comparison, certain specific basic properties should be shown. Of particular importance is whether the divergence/distance etc is invariant to the representation of the graph. For example, If I am comparing graph G_s and G_t, would the resulting number/distance be invariant to vertex label permutations? I.e. distance(G_s, G_t) = distance(permutation1(G_s), permutation2(G_t))? I think this property is very important since it makes sures that the distance is comparing graph structure, not some arbitrary label/representation. As of now it is not clear whether the RGWD satisfies this property. If the property is satisfied, then it would be great to state it explicitly. If it is not satisfied, then at least the authors could mention/discuss it.

Minor Points:

There are some related work/references on comparing graphs of different sizes/number of nodes with no alignments that I think the authors could cite:

Network Portrait Divergence:
An information-theoretic, all-scales approach to comparing networks. James Bagrow and Erik Bollt

Embedded Laplacian Distance:
Multiscale Graph Comparison via the Embedded Laplacian Distance. Edric Tam, David Dunson

Graph Diffusion Wasserstein Distance:
Barbe et. al. Graph Diffusion Wasserstein Distances. ECML-PKDD 2020

**Summary Of The Paper:**

The paper introduces a robust approach for dictionary learning on graphs. The method is based on a robust version of the Gromov-Wasserstein discrepancy, which involves a minimax optimization problem that is neither convex/concave. The authors proved several properties of the RGWD. The authors demonstrate the efficacy of their approach via experiments and simulations.

**Summary Of The Review:**

I think overall the paper is well written, the approach is natural, and the contributions are sufficiently novel/original. However, there remains some areas for improvement that I think if addressed, would make the paper much stronger. In particular, if they could compare against more alternatives in their simulations, add the very related references that are suggested, and comment on whether their method satisfies the permutation invariance property which is very important and natural.

---

> ### Author Response · Authors · 2022-11-17
> **Responses to Reviewer q8cY**
>
> Thank you for taking the time to review our paper and for the invaluable comments.
>
> **More comparisons on simulated datasets.**
> We have further tested state-of-the-art methods including GWF and SC for simulated datasets.
> As is demonstrated in Sec. 5.1 of our revision, our method is superior to a broad set of competitors.
>
> **Invariance to the permutation of the graph.**
> RGWD is invariant to the permutation of the graphs, since the optimal transport plan is also permuted according to the change of the order of nodes.
> We have included this result in Theorem 1 and provided the proof in the Appendix.
>
> **More related work.**
> We have enhanced the discussion in Sec. 6 and incorporated these methods.

---

> > ### Comment · Reviewer_q8cY · 2022-11-17
> > **Updated Score**
> >
> > In light of the author's response, which addressed my concerns comprehensively, I am bumping my score for this paper up to a "accept", which is 8.

---

### Author Response · Authors · 2022-11-17
**Thanks to All Reviewers**

We would like to thank all reviewers for their invaluable comments.
Following these comments, we made minor revisions to our paper and uploaded a new version.
Note that the main paper and the appendix are combined into the same PDF file.
As per the guidance of ICLR-2023, the *pdfdiff* tool of OpenReview can be applied to compare new changes of the paper against the original submission.

---

### Decision · Program_Chairs · 2023-01-20

**Decision:**

Accept: poster

**Justification For Why Not Higher Score:**

The running time is cubic, and it is not clear if these methods, or even GDL in general, can have broader applicability.

**Justification For Why Not Lower Score:**

This is a nice paper that articulates an important problem with graph dictionary learning and provides a natural solution.

**Metareview: Summary, Strengths And Weaknesses:**

Vincent-Cuaz et al. recently extended dictionary learning to graphs (GDL), where the topology of each graph is described with a pairwise relation matrix which are compared via the Gromov-Wasserstein discrepancy (GWD). A serious limitation of graph dictionary learning is the lack of robustness. This paper introduces a new scheme based on a robust variant of GWD which involves a minimax optimization problem that is neither convex nor concave. They establish some basic properties (e.g. triangle inequality). Finally they evaluate it experimentally, and indeed it outperforms ordinary GDL in a variety of settings.

**Note From Pc:**

if the above contains the word "oral" or "spotlight" please see: "oral" presentation means -> notable-top-5% and "spotlight" means -> notable-top-25%. As stated in our emails, we are disassociating presentation type from AC recommendations